# Inventory of anthropogenic methane emissions in Mainland China from 1980 to 2010

S. S. Peng[1*], S. L. Piao[1], P. Bousquet[2], P. Ciais[1,2], B. G. Li[1], X. Lin[2], S. Tao[1], Z. P. Wang[3], Y. Zhang[1], F. Zhou[1]

5    [1]Sino-French Institute for Earth System Science, College of Urban and Environmental Sciences, Peking University, Beijing 100871, China
[2]Laboratoire des Sciences du Climat et de l'Environnement, LSCE/IPSL, CEA-CNRS-UVSQ, Université Paris-Saclay, F-91191 Gif-sur-Yvette, France
[3]State Key Laboratory of Vegetation and Environmental Change, Institute of Botany, Chinese Academy of Sciences,
10   Nanxincun 20, Xiangshan, Beijing 100093, China

*Correspondence to*: S. S. Peng (speng@pku.edu.cn)

**Abstract.** Methane ($CH_4$) has a 28-fold greater global warming potential than $CO_2$ over one hundred years. Atmospheric $CH_4$ concentration has tripled since 1750. Anthropogenic $CH_4$ emissions from China have been growing rapidly in the past decades, and contribute more than 10% of global anthropogenic $CH_4$ emissions with large uncertainties in existing global inventories, generally limited to country-scale statistics. To date, a long-term $CH_4$ emissions inventory including the major sources sectors and based on province-level emission factors is still lacking. In this study, we produced a detailed annual bottom-up inventory of anthropogenic $CH_4$ emissions from the eight major source sectors in China for the period 1980-2010. In the past three decades, the total $CH_4$ emissions increased from 24.4 [18.6-30.5] Tg $CH_4$ yr$^{-1}$ in 1980 (mean [minimum-maximum of 95% confidence interval]) to 44.9 [36.6-56.4] Tg $CH_4$ yr$^{-1}$ in 2010. Most of this increase took place in the 2000s decade with averaged yearly emissions of 38.5 [30.6-48.3] Tg $CH_4$ yr$^{-1}$. This fast increase of the total $CH_4$ emissions after 2000 is mainly driven by $CH_4$ emissions from coal exploitation. The largest contribution to total $CH_4$ emissions also shifted from rice cultivation in 1980 to coal exploitation in 2010. The total emissions inferred in this work compare well with the EPA inventory but appear to be 36% and 18% lower than the EDGAR4.2 inventory and the estimates using the same method but IPCC default emission factors, respectively. The uncertainty of our inventory is investigated using emissions factors collected from state-of-the-art published literatures. We also distributed province-scale emissions into 0.1° x 0.1° maps using social-economic activity data. This new inventory could help understanding $CH_4$ budgets at regional scale and guiding $CH_4$ mitigation policies in China.

## 1 Introduction

Methane ($CH_4$) plays an important role on global warming as a greenhouse gas. The radiative forcing in 2011 relative to 1750
30   caused by anthropogenic $CH_4$ emissions is about 0.97 [0.74-1.20] W m$^{-2}$, ranging from 0.74 to 1.20 W m$^{-2}$, which contributes 32% of total anthropogenic radiative forcing by long-lived greenhouses gases ($CO_2$, $CH_4$, Halocarbons and $N_2O$) since 1750

(Ciais et al., 2013). Atmospheric $CH_4$ concentration increased by 1080 ppb since pre-industrial times, reaching 1803 ppb in 2011 (Ciais et al., 2013). The growth of $CH_4$ levels in the atmosphere is largely driven by increasing anthropogenic emissions (e.g., Ghosh et al., 2015). Based on an ensemble of top-down and bottom-up studies, Kirschke et al. (2013) synthetized decadal natural and anthropogenic $CH_4$ sources for the past three decades, and reported that 50% - 65% of $CH_4$ emissions originate

from anthropogenic $CH_4$ sources.

Between 14% and 22% of global anthropogenic $CH_4$ emissions in the 2000s were attributed to China (Kirschke et al., 2013). The major anthropogenic $CH_4$ sources in China include rice cultivation, fossil fuel exploitation and combustion, livestock, biomass and biofuel burning, and waste deposits. With rapid growth of the Chinese economy, the number of livestock has

nearly tripled in the past three decades, causing an increase in $CH_4$ emissions from enteric fermentation and manure management (Khalil et al., 1993; Verburg and Denier van der Gon, 2001; Yamaji et al., 2003; Zhang and Chen, 2014). The types of livestock (cow, cattle etc.) and their alimentation have evolved as well, and change $CH_4$ emissions (IPCC, 2006). The fossil fuels exploitation and consumption have increased exponentially, especially coal exploitation (e.g., Zhang et al., 2014), although large uncertainties remain in the magnitude of greenhouse gas emissions (e.g. Liu et al., 2015). On the other hand,

the decrease of rice cultivation area (Verburg et al., 2001; Li et al., 2002; Kai et al., 2011) and changes in agricultural practices (Chen et al., 2013) can lead to reduced $CH_4$ emissions from rice paddies.

Total methane emissions from China remain uncertain as illustrated by discrepancies between global inventories, and between bottom-up inventories and recent atmospheric-based analyses (e.g. Kirschke et al., 2013). The Emission Database for Global

Atmospheric Center (EDGAR, version 4.2, http://edgar.jrc.ec.europa.eu/overview.php?v=42) reports that China has 73 Tg $CH_4$ $yr^{-1}$ of anthropogenic $CH_4$ sources in 2008, while U.S. Environmental Protection Agency (EPA) estimates that China emitted 44 Tg $CH_4$ $yr^{-1}$ of anthropogenic $CH_4$ sources in 2010. Based on a province-level inventory, Zhang and Chen (2011) reported anthropogenic $CH_4$ emissions of 38.6 Tg $CH_4$ $yr^{-1}$ for the year 2007. This large range of estimates (~30 Tg $CH_4$ $yr^{-1}$) is mainly caused by different emission factors (EFs) or activity data applied in these inventories (EDAGRv4.2; EPA, 2012;

Zhang and Chen, 2011). Such discrepancies between inventories have been identified as limiting our ability to close the global methane budget (Dlugokencky et al., 2011; Kirschke et al., 2013; Ciais et al., 2013). Atmospheric inversions also tend to infer smaller methane emissions for China than reported by EDGAR4.2, with 59 [49-88] Tg $CH_4$ $yr^{-1}$ for the 2000-2009 decade in Kirschke et al. (2013) and ~40 [35-50] Tg $CH_4$ $yr^{-1}$ in the inversion of Bergamaschi et al. (2013, see their Figure 5).

Global inventories generally rely on country-level socio-economic statistics, which hardly fully reflect the more local to regional, possibly rapidly changing, characteristics of methane sources. This is especially the case in China where economic growth and the sources of $CH_4$ present large differences between provinces. To reduce uncertainties on estimates of Chinese methane emissions, it is therefore of particular importance to build a long-term consistent annual inventory of $CH_4$ emissions for each source sector based on local to regional specific EFs and activity data. This is the main goal of this study.

A comprehensive annual anthropogenic CH$_4$ inventory for Mainland China (PKU-CH$_4$; Note that only 31 inland provinces are included in this inventory, and emissions in Hong Kong, Macau and Taiwan are not included in this inventory) was produced between 1980 and 2010, both at country and province scale, and downscaled at 0.1° spatial resolution. To do so, we compiled activity data at county or province levels for eight major source sectors: 1) livestock, 2) rice cultivation, 3) biomass and biofuel burning, 4) coal exploitation, 5) oil and natural gas systems, 6) fossil fuels combustion, 7) landfills and 8) wastewater. We also compiled regional specific EFs for each source sector from published literature in English and Chinese. We then estimated annual CH$_4$ emissions and their uncertainty for the eight major source sectors and for total emissions. Finally, we produced annual gridded maps of CH$_4$ emissions at 0.1°x0.1° for each source sector based on social-economics drivers (e.g., rural and urban population, coal exploitation, and Gross Domestic Product (GDP)). Note that this inventory only includes annual anthropogenic CH$_4$ emissions, and have not included the seasonality of CH$_4$ emissions yet, which is worth being investigated in future study. The database is described in section 2, methane emissions for the period 1980-2010 are presented in section 3 and discussed in section 4. For the main CH$_4$ source sectors such as emissions from coal exploitation, oil and gas systems, livestock and landfills, the possible reduction potentials and corresponding policies are also discussed in section 4.

## 2 Methods and Datasets

### 2.1 Methodology

The CH$_4$ emissions from 8 sectors, namely livestock, rice cultivation, biomass and biofuel burning, coal exploitation, oil and natural gas systems, fossil fuels combustion, landfills and wastewater are investigated in this study. The methods of IPCC greenhouse gas inventory guidelines (IPCC, 2006) were used to estimate CH$_4$ emissions for these eight sectors. The annual CH$_4$ emissions at the year t from the eight sectors are calculated by Eq. (1).

$$E(t) = \sum_S \sum_R \sum_C AD_{S,R,C}(t) \times EF_{S,R,C}(t) \times (1 - CF_{S,R,C}(t)) \ , \quad (1)$$

Where E($t$) represents the total CH$_4$ emissions from the eight sectors; S, R, and C indicate the index of sectors, regions/provinces and conditions, respectively; AD$_{S,R,C}$($t$) is the activity data at the year t, and EF$_{S,R,C}$($t$) is the emission factor at the year $t$ for sector S, region R and condition C. CF$_{S,R,C}$($t$) is the correction factor at the year $t$ for sector S, region R and condition C, which indicates the fraction of CH$_4$ utilized or oxidized without being released to atmosphere, such as CH$_4$ recovery instead of venting into the atmosphere in coal mining, CH$_4$ oxidation from waste, or reduced emissions due to biogas utilization, for instance. For estimation of CH$_4$ emissions from each source sector, the details of AD$_{S,R,C}$, EF$_{S,R,C}$ and CF$_{S,R,C}$ are introduced in the following Section 2.2. We also applied the same activity data and correction factors but using IPCC default EFs (Table S2) to illustrate the impact of the new EF used in this study compared to the IPCC values. Note that the EFs used in this study do not evolve with time because of the limited information available about time evolution of EFs, which is a limitation of our study.

## 2.2 Activity data, EFs and correction factors

### 2.2.1 Livestock

$CH_4$ emissions from livestock are estimated as the sum of $CH_4$ emissions from enteric fermentation and manure management. Province-level annual census data of domestic livestock for each livestock category were collected from agriculture statistics yearbooks (CASY, 1980-2010). Livestock includes ruminants such as cattle, dairy cattle, buffalo, sheep, and goats, non-ruminant herbivores such as horses, asses, and mules, and omnivorous swine. Because seasonal births and slaughters change the population of livestock, we used slaughtered population and live population at the end of the year to estimate the total emissions from enteric fermentation. Here, average life spans in one year are 12 months for dairy cattle, 10 months for non-dairy cattle and buffalo, 7 months for sheep and goats and 6 months for swine, respectively. The EFs of enteric fermentation and manure management for each category livestock are from published studies are listed in Table 1 (IPCC, 1996, 2006; Dong et al., 2004; Khalil et al., 1993; Verburg and Denier van der Gon, 2001; Yamaji et al., 2003; Zhou et al., 2007). The mean, minimum and maximum of EFs for enteric fermentation from these reported values are summarized in Table 1. For each category of livestock, separated EFs for female, youth and the rest of animals are reported when available.

Because EFs of manure management is a function of mean annual temperature under some special practice (IPCC, 2006), the EFs of manure management from default IPCC (2006) are assigned based on the mean annual temperature for each province (Table 2). The uncertainty of $CH_4$ emissions are estimated by the range of EFs for enteric fermentation and manure management (Table 2) (IPCC, 2006). The $CH_4$ from manure management could be utilized by bio-digester on large scale in China since the 1970s, but there is limited information about $CH_4$ collected from bio-digesters only from manure. We collected the total $CH_4$ emission from bio-digesters with mixed crop straw, manure and waste during the period 1996-2010 from Feng et al. (2012). Before 1996, the annual output of biogas (i.e. avoided $CH_4$ emissions compared to standard manure management practice) was assumed to linearly increase from the early 1980s to 1996, based on the number of household bio-digesters that increased from 4 million in the early 1980s to 6 million in 1996 (Figure S1). Since the biogas contained $CH_4$ from both manure and crop residues, it is assumed that 10%, 15% and 25% of the biogas are low, medium, and high mitigation scenarios for $CH_4$ emissions only from manure management, respectively (Yin, 2015, master thesis), which is removed from the total emissions from standard emissions from manure management in livestock sector. $CH_4$ recovery and reduced emissions due to biogas utilization with manure feedstock is thus accounted in the livestock sector.

### 2.2.2 Rice cultivation

$CH_4$ emissions from rice cultivation sector are estimated using the methodology of Yan et al. (2003). Province-level annual rice cultivation areas (early rice, middle rice and late rice) are collected from agriculture statistics yearbooks (CASY, 1980-2010). The EFs for early rice, middle rice and late rice in five regions under four different cultural conditions (with/without organic input, intermittent irrigation/continuous flooding conditions) are collected from Yan et al. (2003), which summarized

204 season-treatment measurements on 23 different sites (see their Table 2). We apply the EFs from Yan et al. (2003) and rice cultivation areas from yearbooks under different conditions from 1980 to 2010 to calculate $CH_4$ emissions from rice cultivation. 66.7% and 33.3% of rice cultivation area for intermittent irrigation and continuous flooding is assumed as in Yan et al. (2003). There is large uncertainty of rice cultivation area receiving organic input (Huang et al., 1998; Cai, 1997; Yan et al., 2003), and

we assumed 50% of rice paddies received organic input in 2000 (30% of rice paddies have crop straw, green manure or compost and 20% of rice paddies have animal and human waste) according Yan et al. (2003). The practices of organic input have been changing with economic development and policy of agriculture and environment, especially with increasing chemical fertilizer input in the 1980s and 1990s (Figure S2). It is assumed that organic matter input to rice paddies linearly decreased with increasing chemical fertilizer input before 2000, and that the fraction of rice paddy with organic input decreased from 85% in

1980 to 50% in 2000 (Figure S2). After 2000, on the one hand, chemical fertilizer kept increasing (Figure S2) but, on the other hand, the practice of returning crop residues and organic fertilizer applications became popularized again because of policy about       sustainable       quality       of       arable       land       and       air       quality       control       in       China (http://www.sdpc.gov.cn/gzdt/201511/t20151125_759543.html), which can be indirectly supported by increasing number of the machines for returning crop residues in the 2000s (from 0.44 million in 2004 to 0.62 million in 2011). The uncertainties of

rice cultivated areas receiving organic input and irrigation conditions are discussed in the section 4.1. The growing days for early, middle, and late rice are 77, 110-130 and 93 days, respectively (Yan et al., 2003). The correction factors are set as 0 for rice cultivation sector, because no $CH_4$ recovery from rice paddies is observed until now. The uncertainty of $CH_4$ emissions from rice cultivation is derived from the range of EFs (Yan et al., 2003).

### 2.2.3 Biomass and biofuel burning

$CH_4$ emissions from biomass and biofuel burning mainly come from burning of firewood and straw in rural households. In our inventory, this sector includes emissions from firewood and crop residues burnt as biofuel in households and from disposed crop residues burnt in the open fields. Province-level firewood consumption are extracted from the China Energy Statistical Yearbook (1980-2007). Because no firewood data is available after 2007 and firewood consumption in China is stable after 2005 (CESY, 2004-2008; Zhang et al., 2009; Zhang et al., 2014), we assumed that the consumption of firewood from 2008 to

2010 is stable and equal to the average of 2005-2007 emissions. For crops residues burning, we distinguish crops residues used as biofuels in the houses from those burnt in open fields, following Tian et al., (2011). The total crop residues are calculated as annual crops yields and straw-grain ratio for major crops (rice, wheat, corn, soy, cotton and canola) in China. The crops residues burning as biomass fuels and disposed fire in open fields are separately calculated by Eq. (2).

$$RB_{crop} = \sum_c R_c \times N_c \times F \times \theta \quad , \quad (2)$$

Where $RB_{crop}$ is the amount of burning crop residues as biomass fuel or disposed fire in open fields (Kg yr$^{-1}$); $c$ is index of crop; $N_c$ is straw-grain ratio for rice (1.0), wheat (1.4), corn (2.0), soy (1.5), cotton (3.0) and canola (3.0); F is the fraction of crop residues used as biomass fuel or disposed fire in open fields (Table 2), which is determined by the province level of economic

development (Tian et al., 2011); $\theta$ is burning efficiency for biomass fuel in households (100%) and fire in open fields (88.9%) (e.g., Cao et al., 2005; Tian et al., 2011).

EFs of $CH_4$ emissions from biomass and biofuel burning were collected from the scientific literature (Zhang et al., 2000; Andreae et al., 2001; Streets et al., 2003; Cao et al., 2008; Tian et al., 2011). We used EFs from firewood of 2.77 ± 1.80 kg $CH_4$ $t^{-1}$ (mean ± standard deviation), and EFs from crop residues for biomass fuel and fire in open fields of 3.62 ± 2.20 kg $CH_4$ $t^{-1}$ and 3.89 ± 2.20 kg $CH_4$ $t^{-1}$, respectively (Tian et al., 2011). The uncertainty of $CH_4$ emissions (95% CI) are estimated from the range of the EFs by 1000 times of bootstrap samples.

**2.2.4 Coal exploitation**

$CH_4$ emissions from coal exploitation include fugitive $CH_4$ from coal mining and post mining. In China, coal exploitation includes both underground and surface coal mines. Generally, $CH_4$ emissions per unit of coal mined from underground is much higher than that from surface (IPCC, 2006). Province-level annual coal production from underground and surface mines were collected from China Energy Statistical Yearbook and China Statistical Yearbook (1980-2010). The EFs of fugitive $CH_4$ from underground and surface mines are significantly different (Zheng et al., 2006; IPCC, 2006; Zhang et al., 2014). Only 5% coal is mined from surface mines on average at country scale, with a fraction of coal mined varying from 0% for most provinces to more than 17% for Inner Mongolia and Yunnan provinces. Here, we calculated $CH_4$ emissions from both underground and surface mines. For $CH_4$ emissions from underground mines, the EFs vary among mines depending on local mines conditions such as depth of mines and methane concentration etc. Zheng et al. (2006) summarized regional EFs from coal exploitation based on measurements from ~600 coal mines in 1994 and 2000, and these regional EFs correlate with properties of regional mines. For example, Southwest of China has higher EFs than other regions, because the coal mines in that region have deeper depth and higher coalbed methane, especially in Chongqing and Guizhou Province (Zheng et al., 2006; NDRC, 2014). We adopted the mean of regional EFs in China are reported in 1994 and 2000 from Zheng et al. (2006) to calculate $CH_4$ emissions from underground coal mining, and the range of the EFs as the uncertainty (Table 2). The EFs of surface coal mines, we adopted the default value (2.5 $m^3$ $t^{-1}$) from IPCC (2006), since there are few measurements of $CH_4$ emissions from surface mines. The EF of $CH_4$ from coal post-mining including emissions during subsequent handling, processing and transportation of coal), is taken as 1.24 $m^3$ $t^{-1}$ (1.18-1.30 $m^3$ $t^{-1}$), according to the weighted average of production from high- and low- $CH_4$ coal mines using IPCC (2006) default EFs for high- (3.0 $m^3$ $t^{-1}$) and low- (0.5 $m^3$ $t^{-1}$) $CH_4$ coal mines (Zheng et al., 2006). Note that $CH_4$ emissions from abandoned mines are not included in our inventory, because 1) abandoned mines are estimated to account for less than 1% of total emissions from coal mining (NRDC, 2014), and 2) the time series of numbers and locations of the abandoned mines are unavailable (NRDC, 2014). In addition, emissions from underground coal fires are not included in our inventory, because 1) it is unclear how much coal is yearly burnt by underground coal fires from 1980 to 2010, and 2) less than 0.01 Tg $CH_4$ $yr^{-1}$ (less than 0.1% of total emissions from coal mining) are emitted from underground coal fires during the 2000s (EDGARv4.2).

Not all CH$_4$ emissions from underground coal mines are released into atmosphere as CH$_4$. A fraction of CH$_4$ from coal mines are collected for flaring or be utilized by coal bed/mine methane in Clean Development Mechanism (CDM) projects (e.g., Bibler et al., 1998; GMI, 2011). The recovery of CH$_4$ from coal mines increased with economic growth and enhancement of coal safety (NDRC, 2014). For example, Zheng et al. (2006) indicates that the recovery of CH$_4$ from coal mines increased from 3.59% in 1994 to 5.21% in 2000. We used the recovery fraction of 3.59% before 1994 and linearly increase from 3.59% in 1994 to 9.26% in 2010 as $CF_{S,R,C}$ in Equation (1). The range of recovery fraction (3.59% - 5.21%) is taken to calculate the uncertainty of CH$_4$ emissions from coal mining. A volumetric mass density of 0.67 kg m$^{-3}$ is used to convert volume of CH$_4$ emission into CH$_4$ mass.

## 2.2.5 Oil and natural gas systems

Province-level annual crude oil and natural gas production were collected from China Statistical Yearbook (1980-2010). The EFs of fugitive CH$_4$ from oil and natural gas systems in China are from Schwietzke et al. (2014a, 2014b), including venting, flaring, exploration, production and upgrading, transport, refining/processing, transmission and storage, as well as distribution networks in this study, which corresponds to definitions of IPCC subcategory 1B2. For the fugitive CH$_4$ from oil systems, the average EF from oil systems is taken as 0.077 kt CH$_4$ PJ$^{-1}$ (2.9 kg CH$_4$ m$^{-3}$ oil), and the uncertainty of EF are 0.058-0.190 kt CH$_4$ PJ$^{-1}$ (2.2-7.2 kg CH$_4$ m$^{-3}$ oil) (see Table 1 in Schwietzke et al., 2014a). For the fugitive CH$_4$ from natural gas systems, the fugitive emissions rates (FER) of natural gas is decreasing from 1980 to 2011 (Schwietzke et al., 2014b). We assumed a FER linear decrease from 4.6% (0.81 kt CH$_4$ PJ$^{-1}$) in 1980 to 2.0% (0.35 kt CH$_4$ PJ$^{-1}$) in 2010, which is today close to the FER (1.9%) in OECD countries in 2010. The range of uncertainty was estimated with a scenario assuming a low FER in China decreasing from 3.9% in 1980 to 1.8% in 2010, and a scenario with high FER in China decreasing from 5.7% in 1980 to 4.9% in 2010.

## 2.2.6 Fossil fuels combustion

Province-level fossil fuels combustion (TJ) were collected from China Energy Statistical Yearbook (1980-2010). We used the default EFs from IPCC (2006) for CH$_4$ emissions from fossil fuels combustion, 1 kg CH$_4$ TJ$^{-1}$ for coal combustion, 3 kg CH$_4$ TJ$^{-1}$ for oil combustion and 1 kg CH$_4$ TJ$^{-1}$ for natural gas combustion, respectively. The uncertainty of the EFs for fuels combustion is 60% (IPCC, 2006).

## 2.2.7 Landfills

Using IPCC (2006), the CH$_4$ emissions from landfills is estimated by First Order Decay (FOD) method as Eq. (3).

$$E_{Landfill}(t) = (1 - e^{-k}) \times \sum_x e^{-k \times (T_L - x)} \times MSW_L(x) \times MCF_T \times F_T \times DOC \times DOC_d \times f * (1 - O_f) \times \frac{16}{12} \quad , \quad (3)$$

Where $E_{landfill}(t)$ is CH$_4$ emissions from landfills at the year $t$; $k$ is reaction constant and $T_L$ is decay lifetime period, which are 0.3 and 4.6 years based on national inventory (NDRC, 2014); $x$ is the year start to count. $MSW_L$ is the total amount of municipal solid waste (MSW) treated by landfills at province scale; $MCF_T$ is methane correction factor, which corrects CH$_4$ emissions from three types of landfills $T$ ($MCF_T = 1.0$ for managed anaerobic landfills; $MCF_T = 0.8$ for deep ($> 5$ m) non-managed landfills, and $MCF_T = 0.4$ for shallow ($< 5$ m) non-managed landfills) (IPCC, 2006; NDRC, 2014). $F_T$ is the fraction of $MSW_L$ for each type landfill. We adopted the values of $F_T$ by investigation for each province (Du, 2006, master thesis), which are shown in Table 2. $DOC$ is fraction of degradable organic carbon in MSW, and is 6.5% in China (Gao et al., 2006). $DOC_d$ is fraction of $DOC$ that can be decomposed; $f$ is fraction of CH$_4$ in gases of landfill gas, and $O_f$ is oxidation factor and is set as 0.1 in this study. We adopted 0.6 for $DOC_d$ and 0.5 for $f$ in this study (Gao et al., 2006).

Country-total amount of MSW were collected from China Statistical Yearbook (1980-2010). Province-level amount of MSW in 1980, 1985-1988, 1996-2010 were collected from China Environmental Statistical Yearbook (1980, 1985-1988, 1996-2010). The missing province-level MSW were interpolated between periods, and the sum of province-level interpolated data keep conserved with country-total from the national yearbook. The amount of MSW treated by landfills are only available after 2003, and the rest MSW are treated compost, combustion and other processes. The fraction of $MSW_L$ linearly decreases with GDP (R$^2$=0.95, P<0.001; Figure S3). We used this linear relationship to get the fractions of $MSW_L$ before 2003, and assumed 1970s have similar $MSW_L$ as the year of 1980. For uncertainty of CH$_4$ emissions from landfills, maximum CH$_4$ emissions with $DOC_d$=0.6 and $f$=0.6 and minimum CH$_4$ emissions with $DOC_d$=0.5 and $f$=0.4 were calculated.

### 2.2.8 Wastewater

CH$_4$ emissions from wastewater (domestic sewage and industrial wastewater) is estimated by Eq. (4).

$$E_{wastewater}(t) = COD(t) \times B_o \times MCF \quad , \quad (4)$$

Where $E_{wastewater}(t)$ is CH$_4$ emissions from wastewater treatment and discharge at the year $t$; COD(t) is the total amount of chemical oxygen demand for wastewater at the year $t$; $B_o$ is maximum CH$_4$ producing capacity, 0.25 kgCH$_4$/kgCOD; MCF is methane correction factor for wastewater. The total CH$_4$ emissions from wastewater include two parts: one part from wastewater treated by wastewater treatment plants (WTPs) and the other part from wastewater discharged into rivers, lakes or ocean. Here, we adopted 0.165 and 0.467 for MCF of domestic sewage and industrial wastewater treated by (WTPs), respectively (NDRC, 2014). For wastewater discharged into rivers, lakes or ocean, we adopted 0.1 for MCF (IPCC, 2006; NDRC, 2014; Ma et al., 2015).

Annual province-level amount of domestic sewage and industrial wastewater treated by WTPs or discharged into rivers, lakes or ocean were collected from China Statistical Yearbook (1998-2010). In the past three decades, China's economy grows with growth of population and the total amount of domestic sewage water exponentially increased with population (Figure S4). The COD in domestic sewage and industrial wastewater treated by WTPs increases with GDP (R$^2$=0.95-0.99, P<0.001; Figure S4

and S5). The fraction of discharged COD from industrial wastewater decreases with GDP (Figure S5). We used these relationship to interpolate the amount of COD in wastewater treated by WTPs and discharged into rivers, lakes or ocean before 1998, then distribute the total amount of COD into each province using the average contribution of each province to the total for the period 1980-1998.

The uncertainty of $CH_4$ emissions from wastewater mainly comes from the MCF term, besides the amount of COD in wastewater (IPCC, 2006; Ma et al., 2015). We assumed maximum $CH_4$ emissions with MCF=0.3 for domestic sewage and MCF=0.5 for industrial wastewater treated by WTPs, and minimum $CH_4$ emissions with MCF=0.1 for domestic sewage and MCF=0.2 for industrial wastewater treated by WTPs (IPCC, 2006; Ma et al., 2015).

**2.3 Maps of CH$_4$ emissions**

In order to produce gridded emissions maps at 0.1°x0.1° for each source sector, we distributed the province-level $CH_4$ emissions using different activity data(Table S1). First, we collected county-level rural population (CSYRE, 2010), gridded total population and GDP with 1km spatial resolution in 2005 and 2010 (Huang et al., 2014), gridded numbers of animals in 2005 (Robinson et al., 2011), gridded harvested area of rice (Monfreda et al., 2008), annual production of 4264 coal production
sites (Liu et al., 2015), and converted/resampled them into 0.1° by 0.1° gridded maps. Then, these gridded maps are applied to distribute the province-level of $CH_4$ emissions from the eight source sectors (Table S1). Because not all proxy data are available for every year during the period 1980-2010, we only used the activity data for 2005 and 2010 (proxy data in 2005 for the years before 2005, and proxy data in 2010 for the years between 2005 and 2010), therefore assuming that the changes in the spatial structures of the gridded maps remain limited.

**3 Results**

**3.1 Total and sectorial CH$_4$ emissions**

Figure 1 shows the evolution of anthropogenic $CH_4$ emissions in China for the eight major source sectors and for the country-
total, and Table 3 lists the magnitude of $CH_4$ emissions and their uncertainty in 1980, 1990, 2000 and 2010. In 1980, the country-total $CH_4$ emissions was 24.4 [18.6-30.5] Tg $CH_4$ yr$^{-1}$ (Table 3). Rice cultivation and livestock contributed 71% of anthropogenic $CH_4$ sources in 1980, followed by coal exploitation (14%) (Figure 1b). In the past 30 years, the $CH_4$ emissions doubled, reaching 44.9 [36.6-56.4] Tg $CH_4$ yr$^{-1}$ in 2010 (Figure 1a). In 2010, coal exploitation became the largest contributor of Chinese $CH_4$ emissions (40%), followed by livestock (25%) and rice cultivation (16%) (Figure 1c). The increase of $CH_4$
emissions between 1980 and 2010 is mainly attributed to coal exploitation (70% of the total increase) mostly after 2000, followed by livestock (26%) mostly before 2000.

Figure 2 shows the evolution of individual $CH_4$ sources from 1980 to 2010. Among the eight major source sectors, $CH_4$ emissions from seven source sectors increased from 68% to 407%, and only $CH_4$ emissions from rice cultivation decreased by 34% (Figure 2) before 2005 because of decreased rice cultivation area in this period. The increase of country-total $CH_4$ sources accelerates after 2002 (from 0.5 Tg $CH_4$ $yr^{-2}$ before 2002 to 1.3 Tg $CH_4$ $yr^{-2}$ after 2002, Figure 2a). The increase of $CH_4$ emissions in the 2000s contributes 63% of the total increase observed between 1980 and 2010 (Table 3). The acceleration of emissions starting from 2002 is mainly driven by coal exploitation (Figure 2a and 2e), while $CH_4$ emissions from livestock, biomass and biofuel burning, landfills and rice cultivation remain stable or increased at a lower rate after 2002 resulting from the stable or slow increase in activities data in these sectors. Although $CH_4$ emissions from oil and gas systems, fossil fuels combustion and wastewater increased exponentially after 2002, they only contributed less than 13% of the increase in total $CH_4$ emissions in the 2000s.

## 3.2 Spatial patterns of $CH_4$ emissions

Figure 3 shows the spatial distributions of $CH_4$ emissions in 2010 (Note that Figure 3a-3i have different color scales). The total emissions of each province in 1980, 1990, 2000 and 2010 are also listed in Table S3. Hotspots of $CH_4$ emissions are distributed mostly in the densely populated area, where we describe the emissions for South, Center and North of China country (Figure S6 shows the map these regions). These hotspots are driven by livestock, rice cultivation and coal exploitation (Figure 3). North of China has high $CH_4$ emissions from livestock, biomass and biofuel burning, coal exploitation, oil and gas systems, landfills and wastewater. South and central of China has high $CH_4$ emissions from rice cultivation, landfills and wastewater (Figure 3c). Southwest of China has high $CH_4$ emissions from rice cultivation and coal exploitation (Figure 3c and 3e). $CH_4$ emissions from biomass and biofuel burning, oil and gas systems, fossil fuels combustion, landfills and wastewater have one order of magnitude smaller than that from livestock, rice cultivation and coal exploitation. $CH_4$ emissions from biomass and biofuel burning are mainly distributed in the north of China. $CH_4$ emissions from landfills and wastewater are mainly distributed in north, northeast and coast of China. $CH_4$ leakages from oil and gas systems are located in the north part of China, where oil and gas are mostly produced (Figure 3f). $CH_4$ emissions from fossil fuels combustion also concentrate in east part of China (Figure 3g and 3i).

Figure 4 shows the spatial distribution of the changes of $CH_4$ emissions from 1980 to 2010. The $CH_4$ emissions increased in most parts of China, except in western China where there is no significant increase, and in South and Southeast of China where total emissions are decreasing (Figure 4a). The decrease in $CH_4$ emissions in South and Southeast of China is attributed to a decline in rice cultivation, livestock and biomass and biofuel burning emissions, which offsets the increase from other sources in these regions (Figure 4). The increase in $CH_4$ emissions in North and Northeast of China are attributed to livestock, biomass and biofuel burning, coal exploitation, landfills and wastewater. Southwest of China has increase in $CH_4$ emissions from coal exploitation and landfills (Figure 4).

## 4 Discussion

### 4.1 Comparison with other inventories

Figure 2 shows the comparison of $CH_4$ emissions inferred in this study with EGDARv4.2 (EDGAR, http://edgar.jrc.ec.europa.eu/overview.php?v=42), EPA (EPA, 2012) inventories and estimates with IPCC default EFs (hereafter called IPCC-EF estimates, Table S2). We also make comparison of the emissions in 2005 in the text and Figure 2 with the Initial (1994) and Second (2005) National Communication on Climate Change (NCCC) of The People 's Republic of China to UNFCCC (SDPC, 2004; NDRC, 2012). Our estimates of the total $CH_4$ emissions are very close to EPA estimates and 30-40% lower than EDGARv4.2 inventory during the period 1980-2008 (Figure 2a). Compared to IPCC-EF values, our estimates are consistent with it before 2000, but ~30% lower after 2000. The $CH_4$ emissions during 2000-2008 from Regional Emission inventory in Asia (REAS, http://www.nies.go.jp/REAS/) are very close to EDGARv4.2 in China (Kurokawa et al., 2013), so we only compared our estimates with EDGARv4.2 to avoid duplicated comparison. Our estimates during the 2000s are also in better agreement with atmospheric inversions for anthropogenic emissions, which consistently infer smaller emissions in China than EDGAR4.2 (e.g. Bergamaschi et al., 2013, Kirschke et al., 2013). Although the magnitude of the total $CH_4$ emissions do not agree between EDGARv4.2, EPA and this study, the trends of the total $CH_4$ emissions from these three estimates are qualitatively similar, confirming the slow increase before 2002 and the acceleration thereafter (Figure 2a). However, the magnitude of the trend of anthropogenic $CH_4$ emissions after 2002 found in this study (1.3 Tg $CH_4$ $yr^{-2}$) and in EPA (0.7 Tg $CH_4$ $yr^{-2}$) are respectively 63% and 80% less than in EDGAR4.2 (3.5 Tg $CH_4$ $yr^{-2}$). This discrepancy is due mostly to coal exploitation (figure 2e) with smaller contributions from landfills (figure 2h) and oil and gas systems (figure 2f). The slower increase of total $CH_4$ emissions in China than reported by EDGARv4.2 has already be noticed (e.g. Bergamaschi et al., 2013; Saunois et al., 2016) and is improved in the new EDGARv4.3.2 release, in which the total fugitive emissions from coal mining in China is 1.6 times lower than EDGARv4.2 and distributed over about 20 times more point source locations. Lin (2016) assessed the EDGARv4.2 and EDGARv4.3.2 coal mine emissions within her inverse modelling study and showed lower coal mine emissions than EDGARv4.2 over Asia.

In the 1980s, compared with our estimate, higher emissions in EDGARv4.2 are attributed to rice cultivation (additional 7.3 Tg $CH_4$ $yr^{-1}$), wastewater (+3.6 Tg $CH_4$ $yr^{-1}$), biomass and biofuel burning (+2.7 Tg $CH_4$ $yr^{-1}$), and coal exploitation (+3.2 Tg $CH_4$ $yr^{-1}$). In the 2000s, higher emissions from EDGARv4.2 are attributed to coal exploitation (+8.7 Tg $CH_4$ $yr^{-1}$), rice cultivation (+6.0 Tg $CH_4$ $yr^{-1}$), wastewater (+3.8 Tg $CH_4$ $yr^{-1}$), landfills (+1.2 Tg $CH_4$ $yr^{-1}$), biomass and biofuel burning (+1.2 Tg $CH_4$ $yr^{-1}$) and oil and gas systems (+0.8 Tg $CH_4$ $yr^{-1}$). EPA estimates of $CH_4$ emissions from most source sectors are in line with our estimates, except for fossil fuels combustion and wastewater (Figure 2f & 2i), due mainly to the discrepancy between local and IPCC default EFs (NDRC, 2014; IPCC, 2006). IPCC estimates are close to our estimates in a majority of source sectors, except for higher values in coal exploitation and lower values in rice cultivation and landfills.

**Livestock**. $CH_4$ emissions from livestock are the only one to be consistent between the four inventories (Figure 2b). Similar magnitudes of livestock emissions (~10 Tg $CH_4$ $yr^{-1}$) are also reported in previous studies (Verburg and Denier van der Gon, 2001; Yamaji et al., 2003; Zhang and Chen, 2014b). Our estimate in 1994 (11.3 Tg $CH_4$ $yr^{-1}$) is close to the value (11.1 Tg $CH_4$ $yr^{-1}$) in the initial NCCC reported to UNFCCC, but our estimate in 2005 (12.4 Tg $CH_4$ $yr^{-1}$) is lower than the value (17.2 Tg $CH_4$ $yr^{-1}$) reported to UNFCCC (NDRC, 2014), which results from higher EFs of enteric fermentation for non-dairy cattle (71 kg $CH_4$ $head^{-1}$ $yr^{-1}$) and dairy cattle (85 kg $CH_4$ $head^{-1}$ $yr^{-1}$) adopted by NDRC (2014). The stagnation of livestock emissions after 2000 is explained by the stable domestic ruminant population (CSY, 2012). The increasing import of livestock products (e.g., meat and milk) may contribute to slow down the increase of domestic livestock population in the 2000s, when the demand of livestock products are increasing in China (http://faostat3.fao.org/). In addition, the uncertainty of activity data could be further investigated by comparison between multiple sources, such as FAO, national statistics and province-level statistics in the future studies. Besides the uncertainty of population, the EF of livestock are highly correlated to the live weight per head (for meat cattle) and milk production per head (for dairy cattle) (Dong et al., 2004; IPCC, 2006). In this study, as in previous studies, we assumed that EF from livestock in China did not evolve with time because of limited information about the weight distribution of each livestock population type besides numbers of animals, although we estimated an uncertainty using different EFs (Table 1). On the one hand, the (unaccounted for) increasing live weight and milk production per head may have increased EFs of enteric fermentation (IPCC, 2006). On the other hand, the increasing share of crop products / crop residues in the diet of livestock may have reduced the EFs of enteric fermentation (Dong et al., 2004). The possible changing EF resulting from increased live weight and milk production per head or more feed with treated crop residues should be investigated in future work.

**Rice cultivation.** Yan et al. (2003) reported 7.8 [5.8-9.6] Tg $CH_4$ $yr^{-1}$ emissions from rice paddies by combining rice cultivation area in 1995 and 204 measurements of $CH_4$ emission rates from rice paddies with/without organic inputs and intermittent irrigation or continuous flooding. The $CH_4$ emissions from rice cultivation in China were reviewed by Chen et al. (2013), who found a similar number, 8.1 [5.2-11.4] Tg $CH_4$ $yr^{-1}$. SDPC (2004) and NDRC (2012, 2014) reported 6.2 Tg $CH_4$ $yr^{-1}$ and 7.9 Tg $CH_4$ $yr^{-1}$ emissions from rice paddies in 1994 and 2005, respectively. This value in 1994 reported by NCCC to UNFCCC is lower than our estimate (8.8 [7.0-10.6] Tg $CH_4$ $yr^{-1}$) in 1994. Our estimates of $CH_4$ emissions from rice paddies (7.3 [5.9-8.8] Tg $CH_4$ $yr^{-1}$ in 2005) is consistent with these previous estimates, while the estimates of EDGARv4.2 (13.2 Tg $CH_4$ $yr^{-1}$ in 2005) is out of the range reported by NDRC (2014), Chen et al. (2013) and our estimates. The large variation of $CH_4$ emission rates from rice paddies in different regions and different management conditions (e.g., organic and chemical fertilizer inputs, straw application and irrigation) can significantly impact the estimates of $CH_4$ emissions from rice paddies (Cai et al., 2000; Zou et al., 2005; Chen et al., 2013). This could be the main reason of the higher estimates in EDGARv4.2 and lower estimates in EPA and IPCC. The uncertainty of the EFs related to rice practices is still large in China. For example, the exact rice cultivation area with irrigation and rain-fed is not reported at national or province level. The area of rice cultivation received crop straw, green manure, compost and chemical fertilizer and the magnitudes of these organic and chemical fertilizer input

are also uncertain (Yan et al., 2003; Chen et al., 2013). But these practices significantly impact the EFs and the total emissions (Huang et al., 1998, 2004; Cai, 2000; Zou et al., 2005). In this study, we assumed that the area of rice with organic input decreased with increasing chemical fertilizer input during the 1980s and the 1990s, and kept constant after 2000 because of both increasing chemical fertilizer input and returning crop residues in the 2000s (Figure S2). Without this assumption, the

trend of $CH_4$ emissions from rice cultivation could be smaller. The area with continuous irrigation may have changed during the past three decades. This could also impact the trend of $CH_4$ emissions from rice cultivation, and need further study to get and analyse detailed irrigation data, if available. A decrease in $CH_4$ emissions from rice cultivation is confirmed in all of these inventories, because 1) the total rice cultivation area is decreasing and 2) rice cultivation moved northward since 1970s (e.g., CASY, 2011; Chen et al., 2013). After 2003, EDGAR4.2 reports a fast increase of rice emissions, which is not found in our

study (figure 2c).

**Biomass and biofuel burning.** For the $CH_4$ emissions from biomass and biofuel burning, EDGARv4.2 has a two-times larger value than EPA and our estimates in the 1980s (Figure 2d). Previous studies reported 1.9-2.4 Tg $CH_4$ yr$^{-1}$ emissions from biomass and biofuel burning by the same method but independent estimates of activities data (SDPC, 2004; NDRC, 2012,

2014; Zhang and Chen, 2014a, 2014b). Tian et al. (2011) conducted emissions inventories of atmospheric pollutants from biomass and biofuel burning during the 2000s in China, and indicated that $CH_4$ emissions from biomass and biofuel burning increased from 1.9 Tg $CH_4$ yr$^{-1}$ in 2000 to 2.2 Tg $CH_4$ yr$^{-1}$ in 2007. Compared to the Global Fire Emission Database (GFED) v4.1 products, our estimates of $CH_4$ emissions from crop residues burnt in the open fields (0.28 [0.05-0.51] Tg $CH_4$ yr$^{-1}$) are larger than so called agricultural fire emissions in GFEDv4.1 (0.09 [0.04-0.18] Tg $CH_4$ yr$^{-1}$). But considering the uncertainty

of distinguishing agricultural fire and wild fire in GFED4.1 products and the poor detection of small agricultural fires using satellites, our estimates are close to the total $CH_4$ emissions including both wild fire and agricultural fire (0.22 Tg CH4/yr) in GFEDv4.1. Most of CH4 emissions from biomass and biofuel burning in China are from firewood and straw burning inside of households (Tian et al., 2011; Zhang and Chen, 2014a). The amount of firewood and straw burning have large uncertainty (Yevich and Logan, 2003; Wang et al., 2013), especially for the time evolution of firewood and straw burning, because they

are not easy to accurately deduce without information about utilization of crop residues during the last three decades when fast urbanization happened. The assumed constant fraction of crop residues burnt in the open fields and in rural household in this study may lead to overestimate $CH_4$ emissions from both firewood and crop residues burning. For improving air quality and reducing aerosol in the air, a ban on burning crop residues in open fields was passed in the late of 2000s. This should further reduce their contribution to $CH_4$ emissions in China. In this study, the $CH_4$ emissions from manure burning in northwest of

China (e.g. Tibetan Plateau) are not accounted in biomass and biofuel burning sector in order to avoid double counting as $CH_4$ emissions from manure management are integrated in the livestock sector. However, the fraction of $CH_4$ emissions from manure burning only account for less than 1% of $CH_4$ emissions from biomass and biofuel burning (Tian et al., 2011).

**Coal exploitation.** Our estimate of $CH_4$ emissions from coal exploitation (see Table 2 and Figure 2e) is consistent with previous studies and reports (e.g., CCCCS, 2000; Zheng et al., 2005; Cheng et al., 2011; SDPC, 2004; NDRC, 2012, 2014; Zhang et al., 2014). For example, $CH_4$ emissions from coal exploitation was estimated of 8.7 Tg $CH_4$ yr$^{-1}$ in 1990 (CCCCS, 2000), 6.5 Tg $CH_4$ yr$^{-1}$ in 2000 (Jiang and Hu, 2005) and 12.2 Tg $CH_4$ yr$^{-1}$ in 2002 (Yuan et al., 2006). SDPC (2004) and

NDRC (2012, 2014) reported 7.1 Tg $CH_4$ yr$^{-1}$ and 12.9 Tg $CH_4$ yr$^{-1}$ emissions from coal exploitation in 1994 and 2005 respectively, which is quite close to our estimate (Figure 2). According to reports of the State Administration of Coal Mine Safety (2008, 2009), $CH_4$ emissions from coal exploitation are 13.8 Tg $CH_4$ yr$^{-1}$ in 2007 and 14.5 Tg $CH_4$ yr$^{-1}$ in 2008, respectively (Cheng et al., 2011). On the one hand, the default EFs of underground coal mines (18 m$^3$ t$^{-1}$ for average, 25 m$^3$ t$^{-1}$ for high- and 10 m$^3$ t$^{-1}$ for low- $CH_4$ coal mines) in IPCC (2006) are higher than the local whole-country-average EFs (21.8

m$^3$ t$^{-1}$ for high- and 4.5 m$^3$ t$^{-1}$ for low- coal mines in Zhang et al., 2014) (e.g., CCCCS, 2000; Zheng et al., 2005; Zhang et al., 2010, 2014). The higher $CH_4$ emissions from coal exploitation in EDGARv4.2 could thus result from their higher EFs of coal exploitation if IPCC default EFs are adopted in EDGARv4.2 (Figure 2e). On the other hand, local EFs vary by regions, because of different depths of coal mines, $CH_4$ concentration and coal seam permeability (e.g., Zheng et al., 2006). These regional EFs of coal mining range from ~20 m$^3$ t$^{-1}$ in southwest of China and ~19 m$^3$ t$^{-1}$ in northeast of China, to ~5 m$^3$ t$^{-1}$ in west, east and

north of China (Table 2; Zheng et al., 2006). The depths of coal mines and coalbed $CH_4$ concentration are regionally variable (Bibler et al., 1998). Regional EFs of coal exploitation should be considered to estimate $CH_4$ emission as we did in this study, resulting in lower estimates of $CH_4$ emissions from coal exploitation than that when applying country-average emission factor (Zhang et al., 2014). The EFs of whole-country-average therefore induces a significant bias to estimate $CH_4$ emissions from coal exploitation (e.g., Zhang et al., 2014). Besides the EFs, the recovery of $CH_4$ from coal exploitation is another key

parameter for estimation of $CH_4$ emissions (e.g., Cheng et al., 2011). This parameter increased from 3.6% in 1994 and to 5.2% in 2000, based upon data of hundreds of individual coal mines (Zheng et al., 2006). In our inventory, we assumed that the recovery of $CH_4$ from coal exploitation kept increasing from 5.2% in 2000 to 9.2% in 2010. This assumption is consistent with the register of validated CBM and CMM projects in China which started from 2004 and increased in 2007/2008 (http://www.cdmpipeline.org/overview.htm, CDM/JI database). The total reduction of $CH_4$ emissions by the implementation

of CBM and CMM in China derived from the CDM/JI pipeline database is ~0.3 Tg $CH_4$ yr$^{-1}$ in 2006 and ~0.9 Tg $CH_4$ yr$^{-1}$ in 2010, which is close to our estimates of increased $CH_4$ recovery in 2006 (0.4 Tg $CH_4$ yr$^{-1}$) and 2010 (0.8 Tg $CH_4$ yr$^{-1}$). On the top of EFs differences, the increased recovery of $CH_4$ from coal exploitation can be an additional reason for the higher value of this source in EDGARv4.2, as we applied this increasing recovery of $CH_4$ in this study although the time evolution of this parameter has large uncertainty.

**Oil and gas systems & fossil fuel combustion**. Our estimates of $CH_4$ leakage from oil and natural gas systems are close to estimates of IPCC, but smaller than EDGARv4.2 and higher than EPA (Figure 2f). Our estimates of $CH_4$ emissions from fossil fuels combustion, are close to estimates of EDGARv4.2 and IPCC, but much smaller than estimates of EPA (Figure 2g). NDRC (2014) reported 0.2 Tg $CH_4$ yr$^{-1}$ leakage from oil and natural gas systems and 0.1 Tg $CH_4$ yr$^{-1}$ emissions from fossil

fuels combustion in 2005, which is consistent with our estimates for emissions from fossil fuels combustion but much smaller than our estimates for leakage from oil and natural gas systems. Zhang et al. (2014) reported 0.7 Tg $CH_4$ yr$^{-1}$ leakage from oil and natural gas systems and 0.1 Tg $CH_4$ yr$^{-1}$ emissions from fossil fuels combustion, which are lower than our estimates. In this study, we assumed the medium, low and high scenarios for EFs of fugitive emissions from oil and gas systems (Schwietzke et al., 2014a, 2014b), and the EFs are consistent with EFs reported in USA and Canada in the 2000s (~2%, Höglund-Isaksson et al., 2015). The EFs from oil and natural gas systems have a large spread, and source attribution to oil or natural gas production is also highly uncertain (Höglund-Isaksson et al., 2015). Changes in the natural gas production and distribution technology may change the EFs from natural gas systems (Höglund-Isaksson et al., 2015). This may partly contribute to the decreased FER in our inventory. The activities data applied in these inventories are from national energy statistic data or other global statistic (e.g., CDIAC, IEA), the difference of which is less than 10% (Liu et al., 2015). Thus, the differences in these inventories could come from the uncertainty of EFs. Unfortunately, there is limited information about leakage measurements from pipelines in China, which could help reduce the uncertainty of EFs.

**Landfills**. Gao et al. (2006) calculated 1.9-3.4 Tg $CH_4$ yr$^{-1}$ emissions from Chinese landfills in 2004, using IPCC (1996) default EFs and Tier 1 mass balance method which is not suggested in IPCC (2006). SDPC (2004) reported $CH_4$ emissions from landfills (1.5 Tg $CH_4$ yr$^{-1}$) in 1994, which is higher than our estimate (1.1 [0.8-1.3] Tg $CH_4$ yr$^{-1}$). NDRC (2014) reported detailed $CH_4$ emissions from landfills in 2005 (2.2 Tg $CH_4$ yr$^{-1}$) using first-order decay method in IPCC (2006) with parameters from inventory of Chinese landfills. These two estimates are consistent with our estimate (Figure 2h and Table 2). Zhang and Chen (2014) reported higher estimates (4.7 Tg $CH_4$ yr$^{-1}$) in 2008, using mas balance method with a higher MCF than this study and NRDC (2014). By first-order decay method of IPCC (2006), Li et al. (2015) calculated 3.3 Tg $CH_4$ yr$^{-1}$ emissions from landfills in 2011, which is the maximum estimates of this study (Figure 2h). $CH_4$ emissions from landfills in EDGARv4.2 are different with EPA and our estimates in the 2000s, and the trends of $CH_4$ emissions from landfills are different between EDGARv4.2, EPA and this study (Figure 2h). EDGARv4.2 shows an exponential increase trend of 5-8% yr$^{-1}$ between 1980 and 2010, while EPA shows a smaller trend (<1% yr$^{-1}$) and this study shows an increase trend of 5-10% yr$^{-1}$ before 2005 and stable emissions after 2005. This is because the fraction of total MSW dumped into landfills decreases with GDP (Figure S3) while MSW is increasingly managed by composting and incineration (CEnSY, 2011). In this study, we considered the amount of MSW managed by landfills and province-level specific fractions of MSW treated by the three types of landfills (Table 2; Du, 2006). Our estimates of $CH_4$ emissions from landfills still shows large uncertainty after 2000 (20%) because of large uncertainty for fraction of degradable organic carbon in MSW, and the anaerobic conditions of different types of landfills.

**Wastewaters**. Both EDGARv4.2 and EPA have 3-4 times higher $CH_4$ emissions from wastewater than our estimates (Figure 2i). SDPC (2004) reported similar value (6.2 Tg $CH_4$ yr$^{-1}$) as EDGARv4.2 and EPA in 1994, which is much higher than our estimate (Figure 2i), but NDRC (2012, 2014) reported 1.6 Tg $CH_4$ yr$^{-1}$ emissions from wastewater in 2005. Zhou et al. (2012) reported 1.3 Tg $CH_4$ yr$^{-1}$ emissions from wastewater in the 2000s. With the same COD data from CEnSY (2005-2010), Ma et

al. (2015) adopted MCF from NDRC (2014) and EFs from IPCC (2006), and they obtained 2.2 Tg $CH_4$ $yr^{-1}$ emissions from wastewater in 2010. All these estimates do not consider the recovery of $CH_4$ from wastewater. However, Wang et al. (2011) and Cai et al. (2015) reported a tiny $CH_4$ emissions (<0.1 Tg $CH_4$ $yr^{-1}$) from WTPs in China, and they argued that most COD in wastewater are not removed by anaerobic biological treatments, but by oxidation exposure in WTPs. This suggests that the $CH_4$ emissions from wastewater could be much lower if most of wastewater is treated by oxidation exposure in WTPs. Our estimates may overestimate $CH_4$ emissions from wastewater, with limited information of the wastewater treatments in Chinese WTPs. EDGARv4.2 and EPA probably adopted a higher MCF value for WTPs or higher discharged COD in wastewater, resulting in a higher CH4 emissions. The total COD in wastewater reported by CEnSY (2000-2010) rather than estimated by population used in this study may better represent total COD in WTPs and discharged into natural aquatic systems. In addition, the MCF values in Equation (4) for WTPs and for natural aquatic systems are the key parameters for estimating $CH_4$ emissions from wastewater, and need more samples in future inventory.

## 4.2 Mitigation of $CH_4$ emissions in China

The total anthropogenic $CH_4$ emission of China is estimated to be 38.5 [30.6-48.3] Tg $CH_4$ $yr^{-1}$ on average for the 2000s decade. This large source (~12% of the global anthropogenic $CH_4$ source) offers mitigation opportunities. In the past decade, China has increased the rates of coal-mine methane (CMM) capture and utilization (Higashi, 2009). An amount of ~4 Tg $CH_4$ $yr^{-1}$ CMM is captured and ~1 Tg $CH_4$ $yr^{-1}$ utilized in 2009 (Brink et al., 2013). Under the framework of CDM, $CH_4$ utilization in Chinese CMM increased (Feng et al., 2012; SNCCCC, 2012). So did emission reductions from manure management and landfills. More than 35 million bio-digesters have been built for $CH_4$ utilization between 1996 and 2010, and capture annually 15 billion $m^3$ biogas (Feng et al., 2012). The fast increased recovery of $CH_4$ in the late of 2000s suggests a possible overestimation of $CH_4$ emissions from coal exploitation and manure management in our estimates, because we assumed a conservative or linearly increased recovery fraction for $CH_4$ from coal mining and manure management (see Section 2.2). In the CDM database, ~0.4 Tg $CH_4$ $yr^{-1}$ landfill gas is utilized in 2010, and most of the projects of landfill gas utilization started from 2007 in China.

The consumption of natural gas has exponentially grown in China (SNCCCC, 2012). The urban population using natural gas from pipeline network has tripled in the 2000s, and the total length of gas pipes construction has doubled in the past five years with fast urbanization in China (CESY, 2014). Between 1980 and 2010, urban population has tripled in China, and may reach 1 billion in 2050 (UN, 2014). On the one hand, $CH_4$ leakage from natural gas distribution networks may increase this sector of $CH_4$ emissions in the coming decades, because of growth of urban population and increase in coverage of natural gas pipes (CESY, 2012). But on the other hand, new pipes will benefit of recent technologies contrary to older European, US, and Russian gas networks. Associated to the decrease of rural population, the substitution of firewood and straw in China by natural gas could reduce $CH_4$ emissions from biomass and biofuel burning. With population growth and sustained GDP continues in the coming decades, the $CH_4$ sources from livestock, MSW and wastewater are predicted to increase (e.g.,

https://www.globalmethane.org/; Ma et al., 2015). $CH_4$ emissions from rice cultivation could keep stable because almost stable rice cultivation area since 2005, but may decrease or increase from northward shift cultivation and changes in managements such as organic input and irrigation etc.

$CH_4$ mitigation provides a co-benefit to reduce greenhouse gases emissions and improve air pollution, and energy supply (Shindell et al., 2011). Thus, China has launched a national policy to reduce open burning of crop residues, which cuts down the pollution emissions as well as $CH_4$ (SNCCCC, 2012). China has also improved $CH_4$ mitigation within the Global Methane Initiative (GMI) and the framework of CDM on $CH_4$ mitigation on coal-mine methane, agriculture and MSW (Higashi, 2009; https://www.globalmethane.org/). All of these elements can contribute to reduce $CH_4$ emissions of China in the coming decades.
A more precise assessment of the reduction potential of Chinese $CH_4$ emissions could be further investigated in future research based on the detailed inventory reported here.

**5 Summary**

We collected province-level activity data of agriculture, energy and waste and emission factors of $CH_4$ from the eight major source sectors in Mainland China, and estimated annual $CH_4$ emissions from each source sector from 1980 to 2010. Our
estimates of $CH_4$ emissions considered regional specific emission factors, activity data, and correction factors as much as possible. In the past decades, the total $CH_4$ emissions increase from 24.4 [18.6-30.5] Tg $CH_4$ $yr^{-1}$ in 1980 to 44.9 [36.6-56.4] Tg $CH_4$ $yr^{-1}$ in 2010. The largest contributor to total $CH_4$ emissions is rice cultivation in 1980, but has been replaced by coal exploitation after year 2005. The increase of $CH_4$ emissions from coal exploitation and livestock drive the increase of total $CH_4$ emissions. We distributed the annual province-level $CH_4$ emissions into 0.1° x 0.1° high-resolution maps for each source
sector using different social-economic data depending on the sector. These maps can be used as input data for atmosphere transport models, top-down inversions and Earth System Models, especially for regional studies. Our results were compared to EDGAR4.2 and EPA inventories. Good general consistency is found with EPA but our estimates is lower by 36% [30-40%] than EDGAR4.2 and shows slower increase in emissions after 2000 as in EPA.

We investigated the uncertainty of $CH_4$ emissions by using different EFs from published literatures. The EFs should evolve with level of development (e.g., technology for wastewater treatment, evolution of cattle types etc.), however, because of limited information about time evolution of EFs, the emission factors used in this study do not evolve with time. This may cause additional uncertainty for the time series of $CH_4$ inventory. Besides the uncertainty on emission factors, the activity data and recovery fraction also have their own uncertainty. For example, there is 5%-10% uncertainty energy consumption data in
China (Liu et al., 2015). The recovery of $CH_4$ has limited information and would increase with technology innovation and economic growth. The uncertainty of activity data and utilization fraction China have not been fully investigated in this study, and should be examined in the future study if more data become available. In addition, because of the limitation of activity

and mitigation data availability on monthly scale, the seasonality of $CH_4$ emissions for each source sector which is also important for the atmospheric chemistry modelling (Shindell et al., 2012), is not investigated in this study. If the detailed monthly activity data and mitigation data for each source sector (see Section 2.2) can be available, the full monthly $CH_4$ emissions inventory database could be built based on the bottom-up method used in this study in future.

**Data availability**

CH4 inventory (PKU-CH4) in this study is publicly available on website (http://inventory.pku.edu.cn/), and the intention is to regularly update it every two or three years.

**Acknowledgements**

This study was supported by the National Natural Science Foundation of China (grant number 41671079). Shushi Peng
acknowledge supports from the 1000-talent Youth Program. P. C. received support from the European Research Council Synergy grant ERC-2013-SyG-610028 IMBALANCE-P.

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

**Tables**

Table 1. Emission factors (EFs) of enteric fermentation collected from literature and summarized mean, min, max of EFs used in this study. The S1-S6 indicate values collected from references list in the bottom.

| | | EFs of Enteric fermentation ($kg\ CH_4\ head^{-1}\ yr^{-1}$) | | | | | | | | |
| --- | --- | --- | --- | --- | --- | --- | --- | --- | --- | --- |
| | | S1 | S2 | S3 | S4 | S5 | S6 | Mean | Min | Max |
| | | *Live* | | | | | | | | |
| Dairy cattle | Mature female | 78 | 68 | 70 | 48 | 44 | 78 | 64 | 44 | 78 |
| | Young (<1 yr) | 39 | 68 | 38 | 48 | 44 | 40 | 46 | 38 | 68 |
| | Other | 52 | 68 | 57 | 48 | 44 | 58 | 54 | 44 | 68 |
| Non-dairy cattle | Mature female | 64 | 47 | 51 | 48 | 44 | 60 | 52 | 44 | 64 |
| | Young (<1 yr) | 32 | 47 | 29 | 48 | 44 | 35 | 39 | 29 | 48 |
| | Other | 66 | 47 | 53 | 48 | 44 | 58 | 53 | 44 | 66 |
| Buffalo | Mature female | 63 | 55 | 68 | 48 | 50 | 88 | 62 | 48 | 88 |
| | Young (<1 yr) | 45 | 55 | 38 | 48 | 50 | 48 | 47 | 38 | 55 |
| | Other | 66 | 55 | 57 | 48 | 50 | 68 | 57 | 48 | 68 |
| Sheep | Mature female | 14 | 5 | 7 | 5 | 5 | 5 | 7 | 5 | 14 |
| | Young (<1 yr) | 7 | 5 | 4 | 5 | 5 | 7 | 6 | 4 | 7 |
| | Other | 9 | 5 | 4 | 5 | 5 | 3 | 5 | 3 | 9 |
| Goats | Mature female | 9 | 5 | 7 | 5 | 5 | 5 | 6 | 5 | 9 |
| | Young (<1 yr) | 4 | 5 | 4 | 5 | 5 | 7 | 5 | 4 | 7 |
| | Other | 5 | 5 | 4 | 5 | 5 | 3 | 4 | 3 | 5 |
| Swine | Not divided | 1 | 1 | 1 | 1 | 1 | 1 | 1 | 1 | 1 |
| | | *Slaughtered* | | | | | | | | |
| Cattle and buffalo | | 58 | 53 | | | | | 55 | 53 | 58 |
| Sheep and goat | | 3 | 5 | | | | | 4 | 3 | 5 |
| Swine | | 3 | 4 | | | | | 3 | 3 | 4 |

S1: Revised IPCC 1996 Guidelines; Dong et al., (2004)
S2: IPCC, 2006
S3: Yamaji et al., 2003
S4: Verburg & Vandergon, 2001
S5: Khalil et al. 1993
S6: Zhou et al. 2007

**Table 2**. The regional specific Emission factors (EFs) or parameters described in Section 2.2. Mean annual temperature (MAT), Emission factors (EFs) of CH4 emissions from manure management, fractions of burning crop residues, EFs of coal mining, and fractions of municipal solid waste treated by landfills ($MSW_L$) into different types of landfills.

| Province | MAT (°C) | Dairy cattle | Non-dairy cattle | Buffalo | Sheep | Goats | Swine | Open burning | biomass fuels | Mean | 1994 | 2000 | Managed Landfills | non-managed landfills with depth > 5 m | non-managed landfills with depth < 5 m |
|---|---|---|---|---|---|---|---|---|---|---|---|---|---|---|---|
| | | EFs of manure management | | | | | | Fraction of burning crop residues | | EFs of coal mining From underground coal mines ($m^3\ t^{-1}$), data from Zheng et al., (2006) | | | Fractions of $MSW_L$ treated by different types of landfills (%); Data from Du (2006) | | |
| Beijing | 11.0 | 10.00 | 1.00 | 1.00 | 0.10 | 0.11 | 2.00 | 0.05 | 0.70 | 5.58 | 4.18 | 6.97 | 49.2 | 38.1 | 12.7 |
| Tianjin | 13.6 | 12.00 | 1.00 | 1.00 | 0.10 | 0.11 | 2.00 | 0.05 | 0.70 | - | - | - | 54.2 | 34.4 | 11.4 |
| Hebei | 9.6 | 9.00 | 1.00 | 1.00 | 0.10 | 0.11 | 2.00 | 0.10 | 0.40 | 5.58 | 4.18 | 6.97 | 41.8 | 43.7 | 14.5 |
| Shanxi | 8.8 | 9.00 | 1.00 | 1.00 | 0.10 | 0.11 | 2.00 | 0.10 | 0.45 | 5.58 | 4.18 | 6.97 | 2.0 | 73.5 | 24.5 |
| Inner Mongolia | 4.0 | 9.00 | 1.00 | 1.00 | 0.10 | 0.11 | 2.00 | 0.05 | 0.40 | 5.99 | 6.00 | 5.97 | 25.6 | 55.8 | 18.6 |
| Liaoning | 7.8 | 9.00 | 1.00 | 1.00 | 0.10 | 0.11 | 2.00 | 0.10 | 0.55 | 13.08 | 11.75 | 14.40 | 23.6 | 57.3 | 19.1 |
| Jilin | 4.7 | 9.00 | 1.00 | 1.00 | 0.10 | 0.11 | 2.00 | 0.20 | 0.30 | 13.08 | 11.75 | 14.40 | 17.4 | 62.0 | 20.6 |
| Heilongjiang | 1.4 | 9.00 | 1.00 | 1.00 | 0.10 | 0.11 | 2.00 | 0.20 | 0.55 | 13.08 | 11.75 | 14.40 | 26.3 | 55.3 | 18.4 |
| Shanghai | 16.5 | 15.00 | 1.00 | 1.00 | 0.10 | 0.11 | 3.00 | 0.20 | 0.20 | - | - | - | 0.9 | 74.3 | 24.8 |
| Jiangsu | 15.2 | 14.00 | 1.00 | 1.00 | 0.10 | 0.11 | 3.00 | 0.05 | 0.80 | 5.84 | 5.46 | 6.22 | 82.1 | 13.4 | 4.5 |
| Zhejiang | 16.3 | 15.00 | 1.00 | 1.00 | 0.10 | 0.11 | 3.00 | 0.20 | 0.45 | 5.84 | 5.46 | 6.22 | 33.7 | 49.7 | 16.6 |
| Anhui | 15.9 | 14.00 | 1.00 | 1.00 | 0.10 | 0.11 | 3.00 | 0.05 | 0.80 | 5.84 | 5.46 | 6.22 | 34.5 | 49.1 | 16.4 |
| Fujian | 18.5 | 17.00 | 1.00 | 1.00 | 0.10 | 0.11 | 4.00 | 0.20 | 0.30 | 5.84 | 5.46 | 6.22 | 36.8 | 47.4 | 15.8 |
| Jiangxi | 18.0 | 17.00 | 1.00 | 1.00 | 0.10 | 0.11 | 4.00 | 0.10 | 0.45 | 5.84 | 5.46 | 6.22 | 24.3 | 56.8 | 18.9 |
| Shandong | 13.5 | 12.00 | 1.00 | 1.00 | 0.10 | 0.11 | 2.00 | 0.10 | 0.45 | 5.58 | 4.18 | 6.97 | 49.5 | 37.9 | 12.6 |
| Henan | 14.6 | 13.00 | 1.00 | 1.00 | 0.10 | 0.11 | 3.00 | 0.10 | 0.30 | 7.51 | 7.19 | 7.83 | 46.5 | 40.1 | 13.4 |
| Hubei | 15.7 | 14.00 | 1.00 | 1.00 | 0.10 | 0.11 | 3.00 | 0.10 | 0.70 | 7.51 | 7.19 | 7.83 | 32.8 | 50.4 | 16.8 |
| Hunan | 16.9 | 15.00 | 1.00 | 2.00 | 0.15 | 0.17 | 3.00 | 0.10 | 0.40 | 7.51 | 7.19 | 7.83 | 62.1 | 28.4 | 9.5 |
| Guangdong | 21.3 | 21.00 | 1.00 | 2.00 | 0.15 | 0.17 | 5.00 | 0.20 | 0.55 | 7.51 | 7.19 | 7.83 | 61.8 | 28.6 | 9.6 |
| Guangxi | 20.4 | 20.00 | 1.00 | 2.00 | 0.15 | 0.17 | 4.00 | 0.10 | 0.45 | 7.51 | 7.19 | 7.83 | 27.8 | 54.1 | 18.1 |
| Hainan | 24.5 | 26.00 | 1.00 | 2.00 | 0.15 | 0.17 | 5.00 | 0.10 | 0.45 | - | - | - | 33.7 | 49.7 | 16.6 |
| Chongqing | 15.9 | 14.00 | 1.00 | 2.00 | 0.15 | 0.17 | 3.00 | 0.10 | 0.70 | 20.35 | 19.02 | 21.68 | 70.2 | 22.3 | 7.5 |
| Sichuan | 9.0 | 9.00 | 1.00 | 2.00 | 0.15 | 0.17 | 2.00 | 0.10 | 0.45 | 20.35 | 19.02 | 21.68 | 46.4 | 40.2 | 13.4 |
| Guizhou | 15.4 | 14.00 | 1.00 | 2.00 | 0.15 | 0.17 | 3.00 | 0.10 | 0.40 | 20.35 | 19.02 | 21.68 | 5.7 | 70.7 | 23.6 |
| Yunnan | 15.4 | 14.00 | 1.00 | 2.00 | 0.15 | 0.17 | 3.00 | 0.10 | 0.20 | 20.35 | 19.02 | 21.68 | 18.9 | 60.8 | 20.3 |
| Tibet | -1.5 | 9.00 | 1.00 | 2.00 | 0.15 | 0.17 | 2.00 | 0.05 | 0.20 | - | - | - | 0.0 | 75.0 | 25.0 |
| Shaanxi | 10.8 | 10.00 | 1.00 | 2.00 | 0.15 | 0.17 | 2.00 | 0.10 | 0.45 | 5.99 | 6.00 | 5.97 | 0.0 | 75.0 | 25.0 |
| Gansu | 5.8 | 9.00 | 1.00 | 2.00 | 0.15 | 0.17 | 2.00 | 0.05 | 0.55 | 5.99 | 6.00 | 5.97 | 25.3 | 56.0 | 18.7 |
| Qinghai | -2.0 | 9.00 | 1.00 | 2.00 | 0.15 | 0.17 | 2.00 | 0.05 | 0.80 | 5.99 | 6.00 | 5.97 | 58.8 | 30.9 | 10.3 |

| | | | | | | | | | | | | | | |
|---|---|---|---|---|---|---|---|---|---|---|---|---|---|---|
| Ningxia | 8.1 | 9.00 | 1.00 | 2.00 | 0.15 | 0.17 | 2.00 | 0.05 | 0.45 | 5.99 | 6.00 | 5.97 | 24.5 | 56.6 | 18.9 |
| Xinjiang | 6.0 | 9.00 | 1.00 | 2.00 | 0.15 | 0.17 | 2.00 | 0.05 | 0.20 | 5.99 | 6.00 | 5.97 | 0.0 | 75.0 | 25.0 |

**Table 3.** Total CH$_4$ emissions from the eight major source sectors and their total in Mainland China in four snapshot years (1980, 1990, 2000 and 2010). Values are given in Tg CH$_4$ yr$^{-1}$ (mean [min-max]).

| | CH$_4$ emissions in China (Tg CH$_4$ yr$^{-1}$) | | | |
|---|---|---|---|---|
| | 1980 | 1990 | 2000 | 2010 |
| Livestock | 6.2 [4.9-7.8] | 8.9 [7.0-11.2] | 12.3 [9.9-15.2] | 11.4 [9.3-13.7] |
| Rice cultivation | 11.2 [9.0-13.4] | 10.0 [7.9-12.0] | 7.8 [6.2-9.4] | 7.4 [6.0-8.8] |
| Biomass and biofuel burning | 1.4 [0.4-2.5] | 1.9 [0.5-3.3] | 1.9 [0.5-3.3] | 2.4 [0.6-4.2] |
| Coal exploitation | 3.4 [3.0-3.7] | 6.8 [6.0-7.5] | 6.0 [5.3-6.7] | 17.7 [16.7-20.3] |
| Oil and gas systems | 0.6 [0.5-1.3] | 0.7 [0.5-1.6] | 0.9 [0.7-2.1] | 1.6 [1.4-4.2] |
| FF combustion | 0.0 [0.0-0.0] | 0.0 [0.0-0.1] | 0.1 [0.0-0.1] | 0.1 [0.0-0.2] |
| Landfills | 0.4 [0.3-0.5] | 0.8 [0.5-1.0] | 1.6 [1.0-1.9] | 2.0 [1.3-2.4] |
| Wastewater | 1.2 [0.6-1.2] | 1.2 [0.7-1.3] | 1.5 [0.8-1.7] | 2.3 [1.2-2.6] |
| Total | 24.4 [18.6-30.5] | 30.3 [23.1-38.0] | 32.0 [24.4-40.3] | 44.9 [36.6-56.4] |

FF: fossil fuels

**Figures**

Figure 1. (a) $CH_4$ emissions from the eight major source sectors during the period 1980-2010 in Mainland China. Pie diagram of $CH_4$ emissions (%) in (b) 1980 and (c) 2010.

5    Figure 2. (a) Annual total anthropogenic $CH_4$ emissions in Mainland China, and (b) – (i) $CH_4$ emissions from different source sectors during the period 1980-2010. The shaded area shows the 95% confidence interval (CI) of our estimates. NCCC indicates the values from the initial and second National Communication of Climate Change (NCCC) of China reported to UNFCCC in 1994 and 2005. IPCC-EF refers to the estimates using the same method but IPCC default emission factors, and 5-95% CI is based on high and low estimates of emission factors. Note that the empty circle indicates projected

10   2010 value in EPA and the emission from fossil fuel combustion in 1994 is not reported in the initial NCCC of China reported to UNFCCC.

Figure 3. Spatial distribution of (a) total anthropogenic $CH_4$ emissions, and (b) – (i) $CH_4$ emissions from different source sectors in Mainland China in 2010. The unit of the colorbar is g $CH_4$ $m^{-2}$ $yr^{-1}$. Note that subplots have different color scale.

Figure 4. Spatial distribution of changes in (a) total anthropogenic $CH_4$ emissions, and (b) – (i) $CH_4$ emissions from different source sectors in Mainland China from 1980 to 2010. The unit of the colorbar is g $CH_4$ $m^{-2}$ $yr^{-1}$. Note that subplots have different color scale.

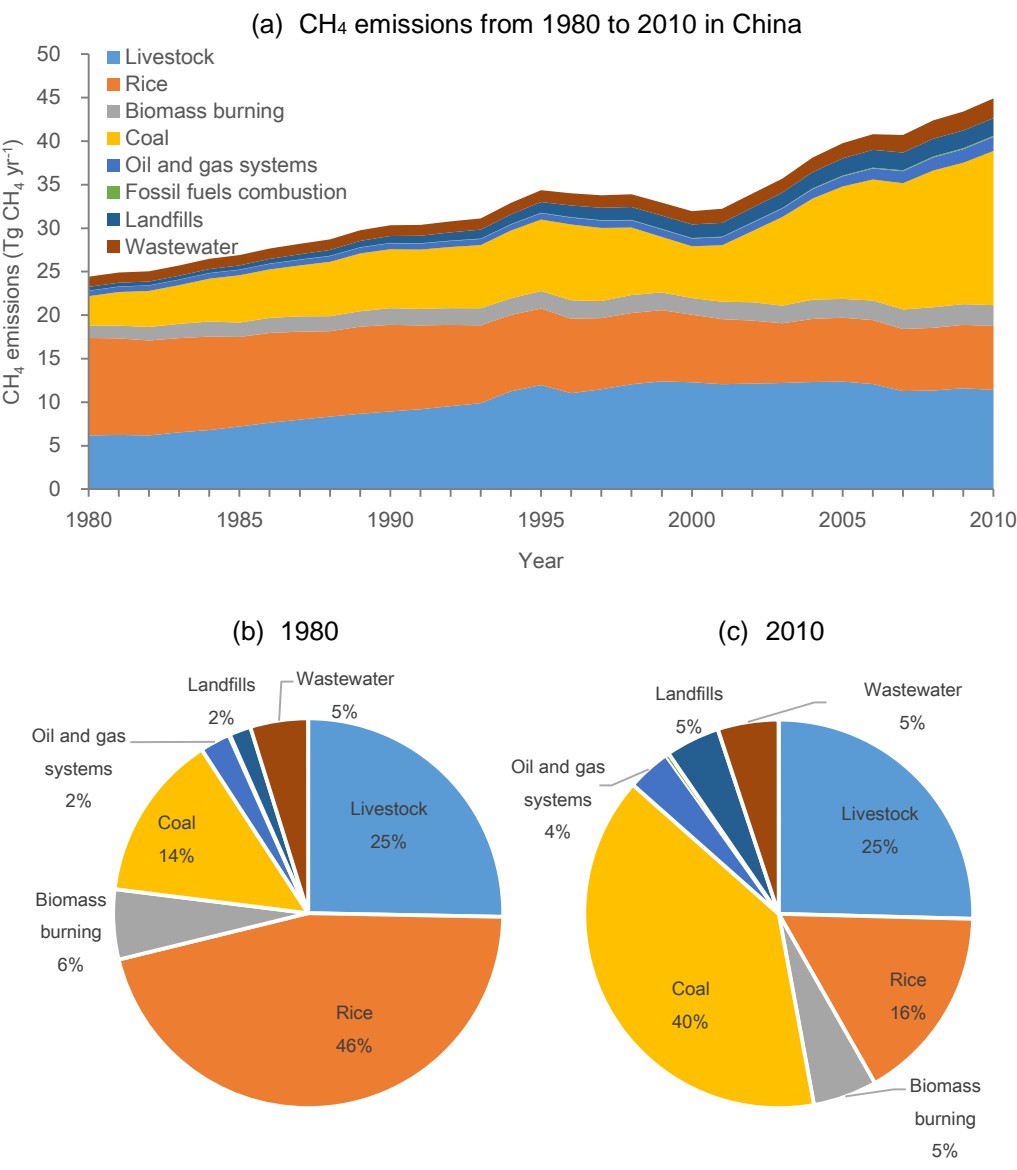

**Figure 1: (a) CH4 emissions from the eight major source sectors during the period 1980-2010 in Mainland China. Pie diagram of CH4 emissions (%) in (b) 1980 and (c) 2010.**

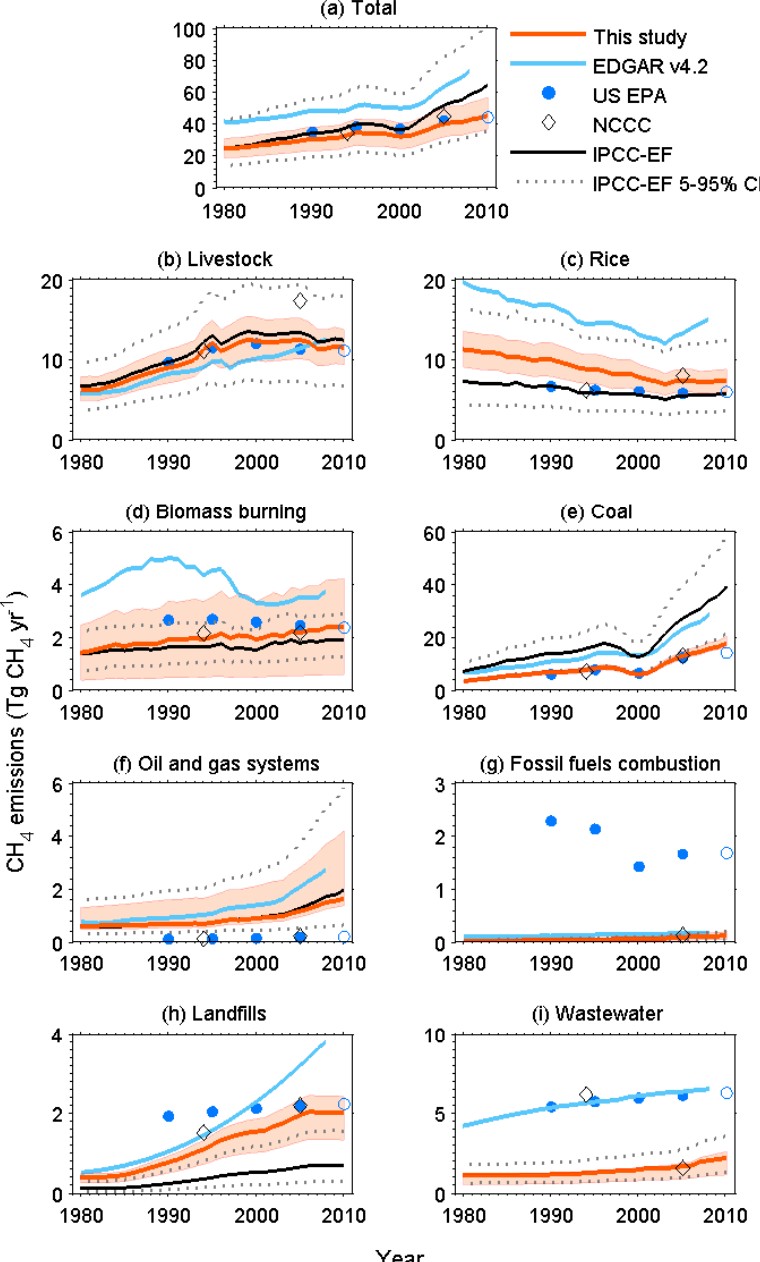

**Figure 2. (a) Annual total anthropogenic CH₄ emissions in Mainland China, and (b) – (i) CH₄ emissions from different source sectors during the period 1980-2010. The shaded area shows the 95% confidence interval (CI) of our estimates. NCCC indicates the values from the initial and second National Communication of Climate Change (NCCC) of China reported to UNFCCC in 1994 and 2005. IPCC-EF refers to the estimates using the same method but IPCC default emission factors, and 5-95% CI is based on high and low estimates of emission factors. Note that the empty circle indicates projected 2010 value in EPA, and the emission from fossil fuel combustion in 1994 is not reported in the initial NCCC of China reported to UNFCCC.**

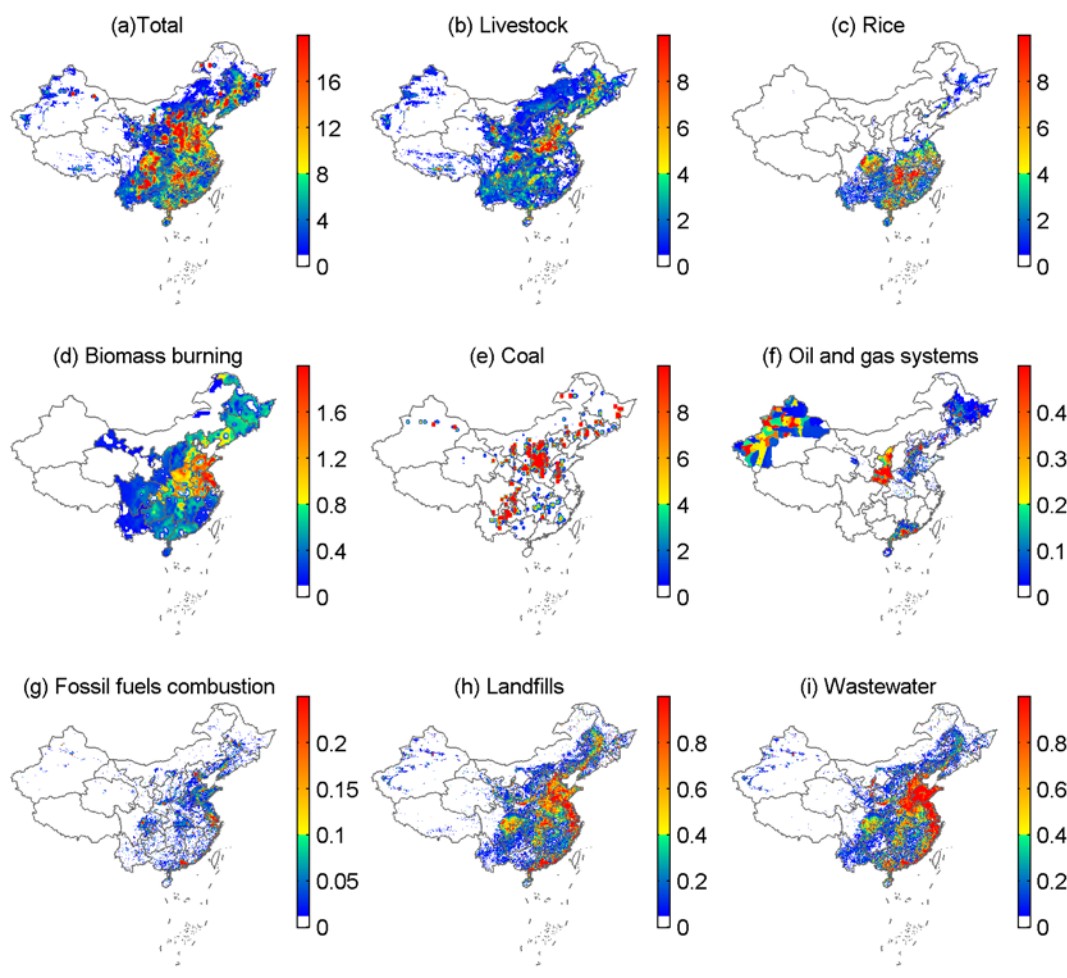

**Figure 3. Spatial distribution of (a) total anthropogenic CH$_4$ emissions, and (b) – (i) CH$_4$ emissions from different source sectors in Mainland China in 2010. The unit of the colorbar is g CH$_4$ m$^{-2}$ yr$^{-1}$. Note that subplots have different color scale.**

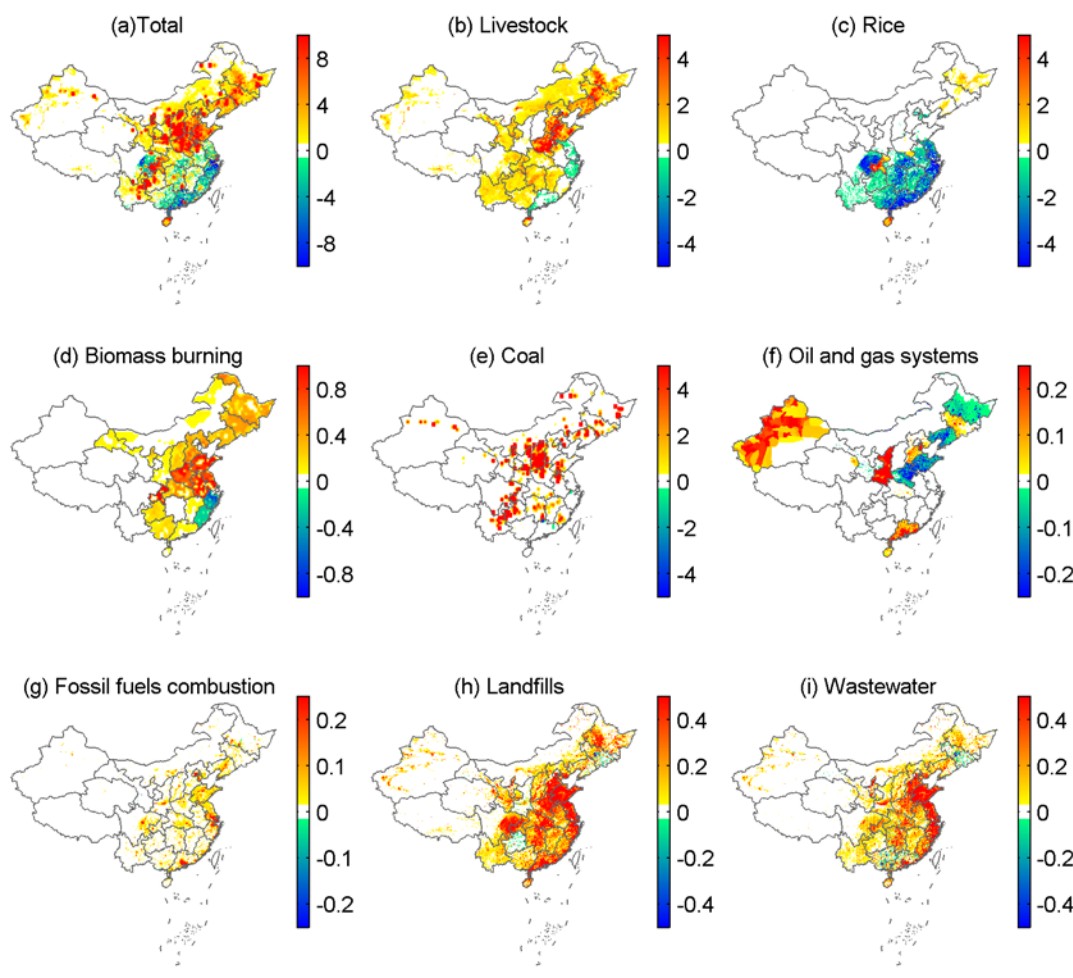

**Figure 4. Spatial distribution of changes in (a) total anthropogenic CH₄ emissions, and (b) – (i) CH₄ emissions from different source sectors in Mainland China from 1980 to 2010. The unit of the colorbar is g CH₄ m⁻² yr⁻¹. Note that subplots have different color scale.**