# Peer review of "Inventory of anthropogenic methane emissions in Mainland China from 1980 to 2010"

_Atmospheric Chemistry and Physics, 2016_

## Referee Comment (RC1) · Anonymous Referee #3 · 13 Apr 2016

Referee comments on:

"Inventory of anthropogenic methane emissions in Mainland China from 1980 to 2010".

2016-04-13

As the authors rightly point out, there are large discrepancies in existing bottom-up inventories of anthropogenic methane emissions for China and there is scope to improve these using more detailed information on a sector level from both international and national sources. In general, I find this paper a thorough contribution in this genre, which adds to existing inventories by the extensive use of data from national Chinese sources, thereby allowing in several sources to go beyond the use of default IPCC emission factors.

I have, however, some concerns on the emission estimations at the sector level, which I would like addressed by the authors. I list them below by sector.

Livestock:

The estimations of CH4 emissions follow standard IPCC methodology, which may explain why emissions fall within a close range of other inventories in Figure 2. I find it somewhat problematic that no adjustment has been made for the use of farm anaerobic digesters in China. I am also surprised that there would be no information available on this when China is since long a world leader on small-scale bio-digesters. I do understand that there is limited information on the number of digesters only digesting manure, since this is not very common. To get reasonable energy efficiency of the digestion, the manure needs to be mixed with at least 20 percent other organic material e.g., straw, food residuals, crop residues. Hence, you would need to look for the number of farm biogas installations that co-digest manure with other organic residues and make assumptions on the fraction of the feedstock that is manure. On p. 15 row 24 you mention that "35 million bio-digesters have been built for CH4 utilization between 1996 and 2010 and capture annually 15 bcm biogas." If the methane content of the biogas is 60%, then it means that 9 bcm CH4 (or about 6 Tg CH4) is captured and utilized annually. Although only some fraction of this can be referred to as methane emissions reduced compared to the practice of not treating manure in digesters, it is still likely to be a significant fraction out of the about 10 Tg CH4 estimated to be released from livestock according to Figure 2. As China is one of few countries with a widespread use of rural small-scale digesters, I find it problematic not at all accounting for this effect on methane emissions.

Rice cultivation:

The estimation of methane emissions from rice cultivation in China based on Yan et al. 2013 is in my opinion state-of-the-art.

Biomass and biofuel burning:

The estimation of methane from this sector draws on information from several national studies and appears robust. I would however like to know how these estimates compare with existing estimates from satellite images of biomass burning e.g., from GFED. To what extent are the estimates consistent/inconsistent?

Coal exploitation:

For this sector, the authors have access to extensive information about depths of coal mines in different provinces as well as the extent of surface mining as opposed to the more common underground mining. This is among the most comprehensive estimates of methane emissions from coal mining in China that I have seen. I have only one question and that is if the authors have been able to assess the prevalence of pre-mining degasification and if the effect of an increasing use of this in China (which is happening according to GMI) in the last decade has been taken into consideration? Is this part of the increased utilization of CH4 from mines that the authors discuss?

Oil and natural gas systems:

I find the emission estimations of this sector the weakest point of the paper and I would like the authors to revise the emission estimations for this sector. The authors claim they are using default emission factors from IPCC (2006), but as shown in the Table below, the emission factor used for oil production is only 15% of the very low end of the IPCC default factor for oil production. For natural gas, the emission factor used is close to the very low end of the IPCC default range. I also include for comparison the corresponding emission factors used by the USA and Canada for their reporting to the UNFCCC. Just like the US and Canada, China's oil and natural gas fields are mostly on-shore and therefore likely to have relatively high emissions from unintended leakage (i.e., fugitive emissions from leakage that are not due to venting of associated petroleum gas (APG)). Moreover, NOAA estimates from satellite images of gas flares that China flares between 2 and 3 bcm of gas annually over the period 1994 to 2010. Most of this gas can be referred to flaring of associated gas primarily from oil production. Although there is not much methane being released from the flaring of associate gas as such, the flaring indicates that there is most likely also venting going on. E.g., for the Canadian province of Alberta, Johnson and Coderre (2011) estimate from measurements that out of total APG generated from conventional oil wells, 97% is recovered for reinjection or utilization, 2.1%

is flared and 0.8% is vented. If we would assume similar circumstances for oil production in China as for conventional oil wells in Canada, it would mean that between 0.76 and 1.1 bcm APG is vented annually from Chinese oil production. If we assume the methane content of APG to be 85% and use the conversion factor 0.7178 kg CH4/m3 CH4, then China would be venting somewhere between 460 and 670 kt CH4 annually from oil production (which is ten times higher than the authors' estimate for 1990, see Table 1). Adding emissions of unintended leakage would increase this number even further. Similar questions can be raised for the emission factor that the authors is using for gas production, transmission and distribution. It seems unreasonably low. Preferable emissions should be estimated separately for gas production, long-distance gas transmission and distribution networks.

Table 1: Methane emission factors for oil and gas systems. Note the emissions factors used in the reviewed paper for China of 0.36 kg/t for oil and 2.77 g/m3 for gas have been converted to kt CH4/PJ to facilitate the comparison.

| | | China Reviewed paper | IPCC (2006) vol.2 Tables 4.2.4 and 4.2.5 | USA NIR (2015) | USA UNFCCC (2014) | | Canada NIR (2015) | | |
|---|---|---|---|---|---|---|---|---|---|
| | | 1990 | Range of default efs kt CH4/PJ produced | 1990 | 2000 | 2010 | 1990 | 2000 | 2010 |
| Oil production (EIA, 2015) | PJ | 6280 | | 16428 | 13003 | 12244 | | | |
| CH4 from oil systems | kt CH4 | 54 | | 1260 | 1496 | 1406.52 | | | |
| Ef oil systems | kt CH4/PJ | 0.009 | Vented associated gas: 0.056-0.39
Flaring associated gas: 0.0001-0.001
Unintended leakage: 0-0.11
Oil refinery: 0.0006-0.0015
**Sum oil systems: 0.057-0.502** | 0.077 | 0.115 | 0.115 | 0.25 (conventional oil)
0.07-0.13 (unconv. oil) | | |
| Natural gas production (EIA, 2015) | PJ | 617 | | 19335 | 20744 | 23007 | | | |
| CH4 from natural gas systems | kt CH4 | 40 | | 7164 | 7459 | 6412.76 | | | |
| Ef gas systems | kt CH4/PJ | 0.065 | Vented associated gas: 0
Flaring associated gas: 0.000005-0.00005
Unintended leakage: 0.01-0.66
Gas transmission & storage: 0.004-0.13
Gas distribution networks: 0.024-0.30
**Sum gas systems: 0.038-1.09** | 0.371 | 0.360 | 0.279 | 0.15-0.16 (gas production) | | |

Finally, on p.9 row 15, authors mention that the province attribution of emissions from oil and gas systems has been done using GDP. There must surely be information available on the geographical distribution of oil and gas production in China. In particular for oil production, almost all emissions are released during extraction and GDP is not likely to be a good measure for the geographical attribution of these emissions.

Fossil fuels combustion:

Use of default IPCC emission factors, which seems appropriate.

Landfills:

Use of FOD method, which is the recommended IPCC method. The levelling off of emissions from landfills towards the end of the period (visible in Figure 2) is explained by an increase in composting and incineration. Estimates seem consistent across mentioned studies.

Wastewater:

Estimates emissions from both domestic and industrial sources. No additional comments.

---

## Referee Comment (RC2) · Anonymous Referee #1 · 14 Apr 2016

**Review of**

**Inventory of anthropogenic methane emissions in Mainland China from 1980 to 2010**

**By Peng et al. (2016), ACPD submitted for publication under ACP**

The paper documents an interesting and unique emissions dataset of methane for China (excluding Hong Kong and Macao) with timeseries 1980-2010 and gridmaps at 0.5degx0.5deg. This CH4 inventory is important input in the first place for the 2 National Communications of 10/12/2014 and of 8/11/2012 of China to UNFCCC but also for the Hemispheric Transport of Air pollution Task Force under the CLRTAP and complements there the MIX dataset, documented in Li et al. (2015, ACPD).

The dataset is a bit weak on:

1) the spatial distribution and could benefit of connecting with Tsinghua University (Q. Zhang) and maybe also with PKU-NH3 (X. Huang) to improve the latter.
2) the temporal resolution which would need to be for the HTAP community at least monthly. The seasonality is in particular important for agricultural sectors, which are the major sectors for CH4. The dataset could improve on this using the temporal profiles in particular for rice cultivation from large literature by Chinese scientists.

The paper compares its inventory with other emissions inventories of USEPA and EDGARv4.2, but should extend this by considering also the national inventories reported by China in its National communications to UNFCCC. The paper also evaluates the changes of the sector-specific emissions over time, but could be completed with a real trend uncertainty and analysis of the major determinants for these trends (such as CH4 recovery of coal mining as pushed under the CDM, change in conditions of rice cultivation, etc.).

*General comments*

The documentation of the dataset could be considerably improved by:

1) Giving a full documentation of the sectors covered (maybe making use of the Common reporting format of the UNFCCC reports) and providing also info on what is not included. E.g. what is included in the gas/oil exploitation? Only gas/oil exploration and venting or also the transmission of gas/oil in pipelines, gas distribution networks (very important source, leading to hotspots in cities). What is not included in the coal exploitation? If the emissions of abandoned mines, closed mines are not estimated, this should be mentioned.
2) Giving a full documentation of the spatial distribution. References for the geo-spatial proxy datasets are missing.
3) Elaborating more on the intercomparison of inventories, including the UNFCCC National Communications of China and using the uncertainty recommendations of IPCC GL (2006)

The content of the paper could be enriched by:

1) Addressing the seasonality, in particular of the agricultural activities. Ideally providing monthly gridmaps with full documentation of used temporal profiles.
2) When describing the emissions at province level, please mention that Macao and Hong Kong are not included. Please compare the emissions magnitude and emission trends between the different provinces. Can there be particular shifts of emissions from one province to another be observed over time? How do the emission factors (per unit of activity) vary amongst the different provinces? Maybe also a mapping of the major emission sectors for each province might be interesting.
3) Highlighting the fact that the database is a fully consistent bottom-up database with activity data and with recovery (correction factor), which allows to conclude for the trend analysis on the determinant factors of some CH4 mitigation measures (e.g. CH4 recovery of coal mining, also CH4 recovering of the gas/oil exploitation, waste separation, …) with the effect they had on the emissions of China. Please derive which reduction potentials further exist.
4) Discussing an outlook on how to maintain and update the database, at which frequency, using which data sources.

*Specific comments*

-) English could be improved: p.1 l13 "have", l14 "contribute"; p4, l18 "are"; p5, l18: remove "emissions", p7, l9 there are few measurements, p12 l26 "and northward of" needs to be corrected; p13, l6, "Yevich", p14, l1 "and 5.2%" should be "to 5.2%"; p.17 l13 "publicly"

-) abstract: please mention that it is an ANNUAL bottom-up inventory

-) page 2 line 22: is the 2010 number of EPA reported/calculated or projected. If it is the latter, please make the difference between reported/calculated data and projected data. Also in fig. 2, make the distinction by have e.g. open circle for projected data.

-) page 3: instead of mentioning "English and Chinese literature", please give the real list of references (and mention the language in the reference list).

-) page 3: formula: what do you mean exactly with "conditions". Do you mean "technologies/ practices, modi operandi"? Moreover: why is the EF not varying in time but only the correction factor?

-) page 4: "CH4 utilization or flaring"? You mean the "CH4 recovery instead of venting into the atmosphere"? Please use the standard reporting language (as also in the CDM)

-) page 4 – Table 1: enteric fermentation is (as described in the IPCC GL (2006)) depending for the dairy cattle on the milk production per head and for the non-dairy cattle and other cattle on the live weight per head. These details would be of interest, also complementing the info in the IPCC GL (2006).

-) page 5: What do you mean exactly with "biomass burning"? Only small scale or also forest fires, etc. ? Moreover, in formula 2: Why do F and theta not have the index C?

-) page 6: in e.g. UK we see huge differences in EF for the fugitive emissions from coal mines, because of different geological underground (based on measurements). Is Zeng et al (2006) for China, a much larger country than UK not reporting a similar large variety?

-) page 6: Have emissions estimates from abandoned mines, closed mines been omitted?

-) page 6 – Table 2: please specify the CH4 recovery of coal mining gas in the table per province. Please add to the Table also the rice cultivation per province and reflecting as such the difference in cultivated area times the number of cropping seasons. This would be valuable information that adds to the information at Chinese province level in the IPCC GL2006)

-) page 7  l7: please specify the EFs in kg CH4 per TJ instead of per kton oil or per m3 gas, because the heat value can change significantly between the different types of oil and different types of gas. Please have an evaluation of the gas distribution leakage. Even though Lelieveld et al (2005, Nature) did not found large leakages from transmission pipelines, it is well-known that the gas distribution networks (especially of the old steel pipeline networks in older cities) are subject to large leakages.

-) page 8, l2: Is the China Env. Stat. Yearbook not showing differences in practices between large versus small or young versus new cities?

-) page 9, l7: please map carefully in a table for each (sub-)sector the specific proxy datasets (over time) are used; page 9, l14 why is livestock distributed with agricultural gross domestic product and GDP and not with the maps of animal numbers, as available from the geonetwork at the FAO site? Why is the oil & gas distributed with GDP, if there are data available on oil and gas exploitation from NOAA? Why considering only 414 coal exploitation sites, if Liu et al (2015, Nature) has a map of several thousand sites. The two-step distribution as described in lines 19-20 should be used for all (sub-)sectors.

-) page 10, l8: please carefully derive when the acceleration in CH4 emissions start, definitely after 2000, but can we even say in 2002 when China joined the WTO?

-) page 12, l16: Seen the relative large variation in rice emissions over time (in EDGARv4.2 varying from 19.2 to 11.9 Tg CH4/yr), please compare the emissions of the same years: so the 2005 value of 13.2 Tg CH4/yr with the NDRC value of 7.9 Tg CH4/yr and with the Chen (2013) estimate of … in 2005.

-) page 13, l1: maybe a discrepancy can be found in the definition of "biomass burning". Please have a careful look what is included: vegetal waste burning, agricultural waste burning, crop residue burning, field burning, grassland fires, woodland fires, forest fires, …?

-) page 14: l3: EDGARv4.2 uses the CDM of UNFCCC as input for all developing countries on coal mine gas recovery (cfr. IEA's CO2 from fuel combustion book, part III, GHG).

-) page 16: please give a quantitative evaluation of the mitigation measures and an outlook on the further reduction potential based on the references. Page 16, l6: please evaluate carefully that new PVC gas distribution networks are better than the old steel networks and that new transmission pipelines (such as for the connection Russia and China) are not expected to lead to high leakages. Input on these issues can be gained also from the Chapter 5 of the AMAP report on CH4 from Hoeglund-Isaksson et al. (2016)

---

## Referee Comment (RC3) · Anonymous Referee #2 · 24 Apr 2016

Review of **S. S. Peng et al.** Inventory of anthropogenic methane emissions in Mainland China from 1980 to 2010, ACP 2016
MS No.: acp-2016-139
MS Type: Research article

The paper provides a consistent time series of CH4 emissions from China from 1980-2010. China is an important contributor to total global CH4 emissions and a better understanding of the sources and possible mitigation options is relevant for the scientific community. Methane emission inventories for China have been made before and as such the work is not novel but the compilation of the different sources and the consistent time series make it certainly worthwhile. Also, as discussed in the paper, the discrepancies between various existing estimates for China is substantial and the investigation of the causes or at least identification of sectors that are most uncertain is important for both the global and the Chinese CH4 budget. I think that for several sources the review of emission factors and especially possible trends in these emission factors or the emission controlling variables over the time period could be more in-depth and that this could still further improve the inventory. On the other hand, an inventory includes many sources and a balance between total time spend on each category and the overall result needs to be found. I would recommend the paper for publication but would like to see several points discussed in more detail or added. If for some reason the authors find it unrealistic or over-demanding to make those changes, some argumentation why this is not feasible or out of scope should be provided.

First of all, as lined out in the beginning of section 2.1., the methods of the IPCC GHG inventory guidelines were followed. The authors then search and use for several, but not all, sources more representative Chinese emission factors. I think it would be valuable to also have a full IPCC emission factor only emission calculation, next to the final result of the paper. This is 1) the easiest way to understand what the impact of the country specific emission factors (EFs) on the total Chinese emission estimate is. 2) In the comparison with the other sources such as EDGAR or EPA – again it would be very useful to know if these estimates' are in line or higher / lower than using avg IPCC EFs. Since the structure followed by the authors is based on the IPCC methodology, my feeling is making an "base-line" avg IPCC EF calculation is not a very demanding task. There are good arguments why the current approach is more accurate but it would provide a very useful benchmark for comparing the impact of more detailed information as well as in the comparison with EDGAR and EPA values.

An important aspect of the paper is the long time series. Something that is not well discussed is whether the activity data and emission factor data really cover the temporal changes. For example if the emission factors are based on using a certain technology but this technology was not used before 1990, the EF might not be representative for the 1980-1990 period. While there are good reasons to use it as best guess, the trend 1980-1990 is then highly uncertain and much less reliable than 1990-2010. I would like to discuss that in more detail for the CH4 emission from rice agriculture.

**CH4 emissions from rice agriculture**

In section 2.2.2 the authors explain their approach to calculate CH4 emissions from rice. While it is clearly acknowledged in the paper that the emission factors depend on such things as organic matter (OM)  input and water management, no trends in these controlling factors are discussed. Denier van der Gon and Neue (1995) and Denier van der Gon (1999) have provided a simple, empirical impact relationship for CH4 emission from rice fields with OM input versus chemical fertilizer. A ~5 t OM/ha input creates a doubling of the CH4 emission, a 10 t OM/ha triples the CH4 emission. Peng et al use an assumption based on Yan et al (2003) that 50% of the rice paddies received organic input. While that may be the case at a certain moment in time for the trend in CH4 emission it is crucial to understand the trend in the OM input because it is such a strong driver of CH4 emissions from rice fields. Denier van der Gon (1999) compiled the green manure statistics, fertilizer production and harvested rice area statistics in China over the period 1960-1995. Especially from the mid-1970s onwards the production of fertilizer in China grows tremendously but the harvested rice area remains the same or declines somewhat. It is a logical hypothesis that the every year increasing availability of fertilizer (urea) started replacing the much more labor-intensive use of OM incorporation. While reliable statistics for total OM use are lacking, the green manure statistics support this hypothesis. From 1980-1990 the harvest rice area slowly declines, the fertilizer production rapidly increases and the planted green manure area roughly halves. The green manure statistics are available at the regional level and show for example a much stronger impact in the Central and east China (See Figure 3 and table 1 in Denier van der Gon, 1999). The impact of less OM input in the rice field is further enhanced by the change of rice varieties from traditional to high yielding varieties. The main trait of these high yielding varieties is that they are very responsive to N fertilizer and allocate (or invest) a much smaller part of their total net primary production in the below ground root system (which will be the OM for the next growing season). This trend is described by Denier van der Gon (2000) but that paper does not  give data for China – nevertheless the high yielding varieties have also been introduced in China and it will also have contributed to making less OM available for CH4 production in Chinese rice soils. This reviewer would therefore argue that the the trend for CH4 from rice as shown in table 3 of the paper, strongly underestimated the trend between 1980 and certainly 1990. An educated guess would be that the year 2000 value is realistic and in line with most available estimates as discussed by the authors but the emissions from rice should show a declining trend in emission since 1980 mostly due to lower OM input into the rice cropping system which is in line with the strong growing availability of urea fertilizer. The authors could use the trends and data compiled in Denier van der Gon 1999 or references therein which would result in a CH4 emission from rice cultivation in an estimated range of ~15 Tg/yr. As a result the trend would be rather similar to EDGAR (Fig 2 in the paper), although the absolute emission level remains lower. Indeed, as mentioned by the authors, the increasing trend in the EDGAR estimate after 2003 is remarkable and not easily understood but that is outside of the scope of the paper.

**CH4 Emissions from Oil and Gas industry**

Emissions from natural gas production sites are characterized by skewed distributions, where a small percentage of sites—commonly labeled super-emitters—account for a majority of emissions. (Zavala-Araiza et al., 2015). The importance of these super-emitters in the O&G sector is a rather new insight and probably not well represented in the current emission factors. It only surfaced due to large numbers of measurements that showed the "fat tail distribution" of the EFs. Therefore, I would argue that using standard emission factors may well lead to underestimation for the emissions from this sector. Moreover, the emission factors used in the paper appear really low. I would like to see a very simple "sanity check" on these numbers. When taking the total calculated CH4 emission from the Oil and Gas industry for example in 2000 (0.1 Tg / yr) or 2010 (0.3 Tg/yr) (see Table 3 in the paper); what share is due to the gas industry and what percentage of total natural gas production is this? And does it make sense over time? At a first glance it seems a really low estimate that is presented here. To get a feeling I have taken the data from Schwietzke et al., 2015 and looked at the CH4 emissions from china from Natural gas industry only if a Fugitive Emission Rate (FER) of 1% is assumed (see figure below). This leads to a factor 2 higher emissions than reported by Peng et al. and the gap is much bigger because in the below estimate oil industry is not included whereas Peng's estimate includes both oil and gas. While this does not mean that the presented estimated in the paper is wrong, I would like to see more discussion and think that expressing the FER as a % of the production is a very useful thing to do to show that really low % are currently assumed in this paper whereas recent measurements in the US and Canada found FER's of 2-4% more realistic.

Constant global avg. Fugitive Emission Release (FER) of 1% of natural gas production only: data taken from Schwietzke et al., 2014. The figure does not include the oil sector emissions yet but these are available from Schwietzke et al and would further increase the emission estimate.

[Figure]

References

Denier van der Gon, H.A.C., Neue, H.U., Influence of organic matter incorporation on the methane emission from a wetland rice field, Global Biogeochemical Cycles 9, pp. 11-22, 1995.

Denier van der Gon, H.A.C., Changes in CH4 Emissions from Rice Fields from 1960 to the 1990s. II. The declining use of organic inputs in rice farming, Global Biogeochemical Cycles 13, 1053-1062, 1999.

Denier van der Gon, H.A.C., Changes in CH4 Emissions from Rice Fields from 1960 to the 1990s. I. Impacts of modern rice technology, Global Biogeochemical Cycles 14, 61-72, 2000.

Sass, R.L., F.M. Fisher, F.T. Turner and M.F. Jund, (1991) Methane Emission from Fice Fields as Influenced by Solar Radiation, Temperature and Straw Incorporation, Global Biogeochem. Cycles, 5, 335-350.

Schwietzke, S.; Griffin, W. M.; Matthews, H. S.; Bruhwiler, L. M. P. Global bottom-up fossil fuel fugitive methane and ethane emissions inventory for atmospheric modeling. ACS Sustainable Chem. Eng. 2014, 2, 1992–2001 DOI: 10.1021/sc500163h.

Zavala-Araiza, D.; Lyon, D.; Alvarez, R. A.; Palacios, V.; Harriss, R.; Lan, X.; Talbot, R.; Hamburg, S. P. Toward a functional definition of methane super-emitters: Application to natural gas production sites. Environ. Sci. Technol. 2015, 49, 8167–8174.

**Data table for figure O&G CH4 emissions derived from Schwietzke et al., 2014:**

year

| | China |
|---|---|
| 1980 | 0.102085 |
| 1981 | 0.090967 |
| 1982 | 0.077018 |
| 1983 | 0.087126 |
| 1984 | 0.088541 |
| 1985 | 0.092382 |
| 1986 | 0.09784 |
| 1987 | 0.099922 |
| 1988 | 0.09923 |
| 1989 | 0.102085 |
| 1990 | 0.102691 |
| 1991 | 0.10633 |
| 1992 | 0.107797 |
| 1993 | 0.112865 |
| 1994 | 0.119005 |
| 1995 | 0.121574 |
| 1996 | 0.143898 |
| 1997 | 0.161981 |
| 1998 | 0.166086 |
| 1999 | 0.179778 |
| 2000 | 0.194534 |
| 2001 | 0.216379 |
| 2002 | 0.233012 |
| 2003 | 0.244863 |
| 2004 | 0.29098 |
| 2005 | 0.356372 |
| 2006 | 0.417752 |
| 2007 | 0.494509 |
| 2008 | 0.542839 |
| 2009 | 0.601392 |
| 2010 | 0.67398 |

---

## Author Comment (AC1) · 28 Aug 2016

Responses to Anonymous Reviewer #1

"Inventory of anthropogenic methane emissions in Mainland China from 1980 to 2010". 2016-04-13

As the authors rightly point out, there are large discrepancies in existing bottom-up inventories of anthropogenic methane emissions for China and there is scope to improve these using more detailed information on a sector level from both international and national sources. In general, I find this paper a thorough contribution in this genre, which adds to existing inventories by the extensive use of data from national Chinese sources, thereby allowing in several sources to go beyond the use of default IPCC emission factors.

**[Response]** Thanks for your careful review and valuable comments.

I have, however, some concerns on the emission estimations at the sector level, which I would like addressed by the authors. I list them below by sector.

Livestock:

The estimations of $CH_4$ emissions follow standard IPCC methodology, which may explain why emissions fall within a close range of other inventories in Figure 2. I find it somewhat problematic that no adjustment has been made for the use of farm anaerobic digesters in China. I am also surprised that there would be no information available on this when China is since long a world leader on small-scale bio-digesters. I do understand that there is limited information on the number of digesters only digesting manure, since this is not very common. To get reasonable energy efficiency of the digestion, the manure needs to be mixed with at least 20 percent other organic material e.g., straw, food residuals, crop residues. Hence, you would need to look for the number of farm biogas installations that co-digest manure with other organic residues and make assumptions on the fraction of the feedstock that is manure. On p. 15 row 24 you mention that "35 million bio- digesters have been built for $CH_4$ utilization between 1996 and 2010 and capture annually 15 bcm biogas." If the methane content of the biogas is 60%, then it means that 9 bcm $CH_4$ (or about 6 Tg $CH_4$) is captured and utilized annually. Although only some fraction of this can be referred to as methane emissions reduced compared to the practice of not treating manure in digesters, it is still likely to be a significant fraction out of the about 10 Tg $CH_4$ estimated to be released from livestock according to Figure 2. As China is one of few countries with a widespread use of rural small-scale digesters, I find it problematic not at all accounting for this effect on methane emissions.

**[Response]** We agree that some of the biogas recuperation could reduce our estimate of $CH_4$ emissions from manure management. The annual output of biogas data is available from 1996 to 2010 (Feng et al., 2012). The number of household bio-digesters increased from 4 million in the early 1980s to 6 million in 1996. If the number of bio-digesters and the annual output of biogas linearly increased from the early of 1980s to 1996, then the annual output of biogas captured increased from 0.4 Tg

CH$_4$/yr in 1980 to 6.2 Tg CH$_4$/yr in 2010 (Figure R1), assuming 60% CH$_4$ in the biogas. However, because the fraction of manure in the mixed organic raw material (mostly mixed with manure and crop residues, or mixed municipal waste) is not clear, several scenarios are needed to estimate how much CH$_4$ emissions from manure management is mitigated by bio-digesters. The CH$_4$ production from manure is about 40% of CH$_4$ production of crop residues in 2012 (Yin, 2015, PhD thesis), and it is assumed that 10%, 15% and 25% of the biogas are low, medium, and high mitigation scenarios for CH$_4$ emissions from manure management, respectively. The biogas reduced CH$_4$ emissions from manure management by 0.1 [0.0-0.1] Tg CH$_4$/yr in 1980 and by 0.9 [0.6-1.6] Tg CH$_4$/yr in 2010. Compared to the CH$_4$ emissions from manure management without biogas mitigation in 2010 (2.3 Tg CH$_4$/yr), the biogas reduced ~40% [27%-68%] of CH$_4$ emissions from manure management in 2010. We added this updated CH$_4$ emissions from livestock with biogas accounted in the revised version.

[Figure]

Figure R1. The annual output of biogas (Tg CH$_4$/yr) from 1980 to 2010. The biogas data is assumed a linear increase from 1980 to 1996. The biogas data from 1996 to 2010 is available from Feng et al. (2012).

Rice cultivation:

The estimation of methane emissions from rice cultivation in China based on Yan et al. 2013 is in my opinion state-of-the-art.

Biomass and biofuel burning:

The estimation of methane from this sector draws on information from several national studies and appears robust. I would however like to know how these estimates compare with existing estimates from satellite images of biomass burning e.g., from GFED. To what extent are the estimates consistent/inconsistent?

**[Response]** We distinguished crops residues used as biofuels in the houses from those burnt in open fields in our inventory. The emissions from fire detected by satellites only include to some extent (detection of small agricultural fires being problematic) the biomass burnt in open fields. In the latest GFED4.1 products, the average $CH_4$ emissions including agricultural fires in China during the period 1997-2010 is 0.09 [0.04-0.18] Tg $CH_4$/yr (http://www.globalfiredata.org/data.html; van der Werf et al., 2010). Our estimation of the average $CH_4$ emissions from crop residues burnt in open fields during the same period is 0.28 [0.05-0.51] Tg $CH_4$/yr, which is higher than that derived from GFED4.1. But considering the uncertainty of distinguishing agricultural fire and wild fire in GFED4.1 products and the poor detection of small fires using burned area from space, our estimates are close to the total $CH_4$ emissions including both wild fire and agricultural fire (0.22 Tg $CH_4$/yr).

Because the changes in biomass burning in open fields is unknown, the fraction of biomass burnt in open fields to total crop residues is assumed to keep constant from 1980 to 2010 in our inventory. The $CH_4$ emissions from biomass burning in open fields in our inventory is increasing with crop residues from 1997 to 2010, but GFED4.1 based on burned area data with fixed emission factors for agricultural fires shows no trend of $CH_4$ emissions. This indicates that the fraction of crop residues burnt in the open fields should be changing in the past two decades, which cause further uncertainty in our inventory. We added this comparison and discussion in the revised version (Page 13, line 11-15).

Coal exploitation:

For this sector, the authors have access to extensive information about depths of coal mines in different provinces as well as the extent of surface mining as opposed to the more common underground mining. This is among the most comprehensive estimates of methane emissions from coal mining in China that I have seen. I have only one question and that is if the authors have been able to assess the prevalence of pre-mining degasification and if the effect of an increasing use of this in China (which is happening according to GMI) in the last decade has been taken into consideration? Is this part of the increased utilization of $CH_4$ from mines that the authors discuss?

**[Response]** Yes, the pre-mining degasification is considered as one way of utilization in our inventory. The increased utilization of $CH_4$ from coal bed methane (CBM) and coal mine methane (CMM) is accounted by the increasing utilization fraction in our study, which increased by 4% in the last decade (from 5.2% in 2000 to 9.2% in 2010). It is assumed the utilization linearly increases from 1980 to 2010 in our inventory. In the CDM/JI pipeline database (http://www.cdmpipeline.org/overview.htm), the registered and validated projects of CBM and CMM in China started from 2004 and increased strongly in 2007/2008. The total reduction of $CH_4$ emissions by CBM and CMM in China derived from CDM/JI pipeline database is ~0.3 Tg $CH_4$/yr in 2006 and ~0.9 Tg $CH_4$/yr in 2010, which is close to our estimates of increased $CH_4$ recovery in 2006 (0.4 Tg $CH_4$/yr) and 2010 (0.8 Tg $CH_4$/yr). We added the discussion about the utilization of CBM and CMM in the revised version (Page 14, line 15-19).

Oil and natural gas systems:

I find the emission estimations of this sector the weakest point of the paper and I would like the authors to revise the emission estimations for this sector. The authors claim they are using default emission factors from IPCC (2006), but as shown in the Table below, the emission factor used for oil production is only 15% of the very low end of the IPCC default factor for oil production. For natural gas, the emission factor used is close to the very low end of the IPCC default range. I also include for comparison the corresponding emission factors used by the USA and Canada for their reporting to the UNFCCC. Just like the US and Canada, China's oil and natural gas fields are mostly on-shore and therefore likely to have relatively high emissions from unintended leakage (i.e., fugitive emissions from leakage that are not due to venting of associated petroleum gas (APG)). Moreover, NOAA estimates from satellite images of gas flares that China flares between 2 and 3 bcm of gas annually over the period

1994 to 2010. Most of this gas can be referred to flaring of associated gas primarily from oil production. Although there is not much methane being released from the flaring of associate gas as such, the flaring indicates that there is most likely also venting going on. E.g., for the Canadian province of Alberta, Johnson and Coderre (2011) estimate from measurements that out of total APG generated from conventional oil wells, 97% is recovered for reinjection or utilization, 2.1% is flared and 0.8% is vented. If we would assume similar circumstances for oil production in China as for conventional oil wells in Canada, it would mean that between 0.76 and 1.1 bcm APG is vented annually from Chinese oil production. If we assume the methane content of APG to be 85% and use the conversion factor 0.7178 kg $CH_4$/m3 $CH_4$, then China would be venting somewhere between 460 and 670 kt $CH_4$ annually from oil production (which is ten times higher than the authors' estimate for 1990, see Table 1). Adding emissions of unintended leakage would increase this number even further. Similar questions can be raised for the emission factor that the authors is using for gas production, transmission and distribution. It seems unreasonably low. Preferable emissions should be estimated separately for gas production, long-distance gas transmission and distribution networks.

Table 1: Methane emission factors for oil and gas systems. Note the emissions factors used in the reviewed paper for China of 0.36 kg/t for oil and 2.77 g/m3 for gas have been converted to kt $CH_4$/PJ to facilitate the comparison.

| | | China | IPCC (2006) vol.2 Tables 4.2.4 and 4.2.5 | USA | | | Canada | | |
|---|---|---|---|---|---|---|---|---|---|
| | | Reviewed paper | | NIR (2015) | UNFCCC (2014) | | NIR (2015) | | |
| | | 1990 | Range of default efs kt $CH_4$/PJ produced | 1990 | 2000 | 2010 | 1990 | 2000 | 2010 |
| Oil production (EIA, 2015) | PJ | 6280 | | 16428 | 13003 | 12244 | | | |
| $CH_4$ from oil systems | kt $CH_4$ | 54 | | 1260 | 1496 | 1406.52 | | | |
| Ef oil systems | kt $CH_4$/PJ | 0.009 | Vented associated gas: 0.056-0.39
Flaring associated gas: 0.0001-0.001
Unintended leakage: 0-0.11
Oil refinery: 0.0006-0.0015
**Sum oil systems: 0.057-0.502** | 0.077 | 0.115 | 0.115 | 0.25 (conventional oil)
0.07-0.13 (unconv. oil) | | |
| Natural gas production (EIA, 2015) | PJ | 617 | | 19335 | 20744 | 23007 | | | |
| $CH_4$ from natural gas systems | kt $CH_4$ | 40 | | 7164 | 7459 | 6412.76 | | | |
| Ef gas systems | kt $CH_4$/PJ | 0.065 | Vented associated gas: 0
Flaring associated gas: 0.000005-0.00005
Unintended leakage: 0.01-0.66
Gas transmission & storage: 0.004-0.13
Gas distribution networks: 0.024-0.30
**Sum gas systems: 0.038-1.09** | 0.371 | 0.360 | 0.279 | 0.15-0.16 (gas production) | | |

Finally, on p.9 row 15, authors mention that the province attribution of emissions from oil

and gas systems has been done using GDP. There must surely be information available on the geographical distribution of oil and gas production in China. In particular for oil production, almost all emissions are released during extraction and GDP is not likely to be a good measure for the geographical attribution of these emissions.

**[Response]** Thank you for carefully checking fugitive emissions from oil and natural gas systems. Compared the default EFs of IPCC (2006) and EFs in Schwietzke et al. (2014a, 2014b), the EFs in the previous version have smaller value. Considering the "realistic" EFs in USA, Canada and other countries from UNFCCC (2014) and Schwietzke et al. (2014a, 2014b) suggested by the reviewer, in the revised version, we adopted the EFs in Schwietzke et al. (2014a, 2014b) for fugitive emissions from oil and natural gas systems. For fugitive emissions from oil systems, the average EF is 0.077 kt $CH_4$/PJ (2.9 kg $CH_4$/m$^3$ oil), and the low and high boundary of EF are 0.058 kt $CH_4$/PJ (2.2 kg $CH_4$/m$^3$ oil) and 0.190 kt $CH_4$/PJ (7.2 kg $CH_4$/m$^3$ oil), respectively (see Table 1 in Schwietzke et al., 2014a). These values of EF are consistent with the EFs in the table listed by the reviewer. The fugitive $CH_4$ emissions from oil systems increase from 0.36 [0.27-0.98] Tg $CH_4$/yr in 1980 to 0.68 [0.52-1.86] Tg $CH_4$/yr in 2010.

For fugitive emissions from gas systems, the fugitive emissions rates (FER) of natural gas is decreasing from 1980 to 2011 (Schwietzke et al., 2014b). In China, we adopted the FER linearly decreases from 4.6% (0.81 kt $CH_4$/PJ) in 1980 to 2.0% (0.35 kt $CH_4$/PJ) in 2010. This medium FER is close to the EF in 2010 in the above table. The low and scenario of FER in China decreases from 3.9% in 1980 to 1.8% in 2010, and the high scenario of FER in China decreases from 5.7% in 1980 to 4.9% in 2010. The fugitive $CH_4$ emissions from gas systems increase from 0.45 [0.38-0.56] Tg $CH_4$/yr in 1980 to 1.27 [1.14-3.11] Tg $CH_4$/yr in 2010.

We agree that GDP is not likely to be a good proxy for the geographical attribution of fugitive emissions from oil and gas systems. In the revised version, we applied the spatial distribution of EDGARv42 grid maps with spatial resolution of 0.1 degree by 0.1 degree, scaled by the total emissions from oil and gas systems in each province (Schwietzke et al., 2014a). The population density, oil and gas production sites, and other proxies for transportation routes are considered in EDGARv42 grid maps for $CH_4$ fugitive emissions from oil and gas systems. Thus, the spatial distributions of $CH_4$ fugitive emissions from oil and gas systems in the revised version include the geographical distribution of oil and gas production in China (Figure 3), which has better geographical distribution than the GDP proxy in the previous version. With all these changes, we think to have deeply revised the emissions from

this sector as requested by the reviewer.

Fossil fuels combustion:

Use of default IPCC emission factors, which seems appropriate.

Landfills:

Use of FOD method, which is the recommended IPCC method. The levelling off of emissions from landfills towards the end of the period (visible in Figure 2) is explained by an increase in composting and incineration. Estimates seem consistent across mentioned studies.

Wastewater:

Estimates emissions from both domestic and industrial sources. No additional comments.

**References**

Schwietzke, S., Griffin, W. M., Matthews, H. S., and Bruhwiler, L. M. P.: Global Bottom-Up Fossil Fuel Fugitive Methane and Ethane Emissions Inventory for Atmospheric Modeling, ACS Sustainable Chemistry & Engineering, 2, 1992-2001, 10.1021/sc500163h, 2014a.

Schwietzke, S., Griffin, W. M., Matthews, H. S., and Bruhwiler, L. M. P.: Natural gas fugitive emissions rates constrained by global atmospheric methane and ethane, Environmental Science and Technology, 48, 7714-7722, 10.1021/es501204c, 2014b.

Van Der Werf, G. R., Randerson, J. T., Giglio, L., Collatz, G. J., Mu, M., Kasibhatla, P. S., Morton, D. C., Defries, R. S., Jin, Y., and Van Leeuwen, T. T.: Global fire emissions and the contribution of deforestation, savanna, forest, agricultural, and peat fires (1997-2009), Atmospheric Chemistry and Physics, 10, 11707-11735, 10.5194/acp-10-11707-2010, 2010.

Yin, D. The research on regional differentiation of rural household biogas in China and the ratio of raw materials. Dissertation for Doctor Degree, Northwest A & F University, 2015.

---

## Author Comment (AC2) · 28 Aug 2016

Responses to Anonymous Reviewer #2

The paper documents an interesting and unique emissions dataset of methane for China (excluding Hong Kong and Macao) with timeseries 1980-2010 and gridmaps at 0.5degx0.5deg. This $CH_4$ inventory is important input in the first place for the 2 National Communications of 10/12/2014 and of 8/11/2012 of China to UNFCCC but also for the Hemispheric Transport of Air pollution Task Force under the CLRTAP and complements there the MIX dataset, documented in Li et al. (2015, ACPD).

The dataset is a bit weak on:

1)          the spatial distribution and could benefit of connecting with Tsinghua University (Q. Zhang) and maybe also with PKU-NH3 (X. Huang) to improve the latter.

2)          the temporal resolution which would need to be for the HTAP community at least monthly. The seasonality is in particular important for agricultural sectors, which are the major sectors for $CH_4$. The dataset could improve on this using the temporal profiles in particular for rice cultivation from large literature by Chinese scientists.

The paper compares its inventory with other emissions inventories of USEPA and EDGARv4.2, but should extend this by considering also the national inventories reported by China in its National communications to UNFCCC. The paper also evaluates the changes of the sector-specific emissions over time, but could be completed with a real trend uncertainty and analysis of the major determinants for these trends (such as $CH_4$ recovery of coal mining as pushed under the CDM, change in conditions of rice cultivation, etc.).

**[Response]** Thanks for your review and valuable comments. We tried our best to improve our inventory of $CH_4$ emissions by 1) with higher spatial resolution (0.1 degree by 0.1 degree) and better proxy for the spatial distribution of $CH_4$ emissions from each sector, and 2) investigating/discussing uncertainty from mitigations such as bio-digesters, coal bed methane (CBM) and coal mine methane (CMM) from registered CDM database, and conditions of rice cultivation, as well as fugitive emissions from oil and gas systems. For the temporal resolution, it is very difficult to estimate the seasonality of $CH_4$ emissions for all the eight sectors because 1) monthly activity data is hardly available, and 2) monthly mitigation data by digesters, CBM and CMM is not available. Thus we targeted to have annual $CH_4$ emissions for all the eight sectors in our inventory, although monthly $CH_4$ emissions from one or two sectors (rice cultivation etc.) could be investigated.

*General comments*

The documentation of the dataset could be considerably improved by:

1)      Giving a full documentation of the sectors covered (maybe making use of the Common reporting format of the UNFCCC reports) and providing also info on what is not included. E.g. what is included in the gas/oil exploitation? Only gas/oil exploration and venting or also the transmission of gas/oil in pipelines, gas distribution networks (very important source, leading to hotspots in cities). What is not included in the coal exploitation? If the emissions of abandoned mines, closed mines are not estimated, this should be mentioned.

**[Response]** Thanks for the reminder. The fugitive emissions from oil and gas systems include emissions from venting, flaring, exploration, production and upgrading, transport, refining/processing, transmission and storage, as well as distribution networks in this study, which corresponds to IPCC subcategory 1B2. For the fugitive $CH_4$ from coal mines, the emissions from abandoned mines are not included in our inventory because of the limitation of unavailable data for abandoned coal mines. Except for the emissions from abandoned coal mines, emissions from mining, post-mining, and flaring defined as IPCC subcategory 1B1 are included in our inventory. We added one sentence for the abandoned mines in the revised version.

2)      Giving a full documentation of the spatial distribution. References for the geo-spatial proxy datasets are missing.

**[Response]** We added the references of the proxy data for the spatial distribution (Table S1). In the revised version, we improved the spatial distribution with higher resolution and better proxy data as the reviewers suggested. For livestock sector, we used spatial distribution of number of animals from global livestock production systems (Robinson et al., 2011) as the proxy data. For the sector of oil and gas systems, we used the proxy data from EDGARv4.2 as Schwietzke et al. (2014a). For the other sectors, we used higher resolution of proxy data. We added these details about spatial distribution in the revised version.

3)      Elaborating more on the intercomparison of inventories, including the UNFCCC National Communications of China and using the uncertainty recommendations of IPCC GL (2006)

**[Response]** We already included the comparison between our inventory and the values in 2005 China reported to UNFCCC (Second National Communication on Climate Change of The

People 's Republic of China (SNCCCC); The National Development and Reform Commission: The People's Republic of China national Greenhouse gas inventory for the year 2005 to UNFCCC, 2014. Beijing, China Environmental Press.). In the revised version, we also included the comparison between our inventory and the values with default IPCC (2006) EFs.

The content of the paper could be enriched by:

1) Addressing the seasonality, in particular of the agricultural activities. Ideally providing monthly gridmaps with full documentation of used temporal profiles.

[Response] It is very difficult to estimate the seasonality of $CH_4$ emissions for all the eight sectors because 1) monthly activity data is hardly available, and 2) monthly mitigation activity data is not available. Thus we targeted to have annual $CH_4$ emissions for all the eight sectors in our inventory, although monthly $CH_4$ emissions from one or two sectors (rice cultivation etc.) could be investigated.

2) When describing the emissions at province level, please mention that Macao and Hong Kong are not included. Please compare the emissions magnitude and emission trends between the different provinces. Can there be particular shifts of emissions from one province to another be observed over time? How do the emission factors (per unit of activity) vary amongst the different provinces? Maybe also a mapping of the major emission sectors for each province might be interesting.

[Response] We already limited our inventory in "Mainland China" in our title, and our inventory only included the provinces in Mainland China. We added one table for the emissions magnitude for each province in the SI (Table S3). The emission factors for each province are shown in Table 2. The spatial patterns of each source sector are already shown in Figure 3.

3) Highlighting the fact that the database is a fully consistent bottom-up database with activity data and with recovery (correction factor), which allows to conclude for the trend analysis on the determinant factors of some $CH_4$ mitigation measures (e.g. $CH_4$ recovery of coal mining, also $CH_4$ recovering of the gas/oil exploitation, waste separation, …) with the effect they had on the emissions of China. Please derive which reduction potentials further exist.

[Response] In this study, we would like to give a full documentation of bottom-up inventory for anthropogenic methane emissions in China. Although the future reduction potentials are

interesting, we would like to focus on the past three decades in this study and simply mention the possible reduction potentials. Quantitatively estimates of the reduction potentials of $CH_4$ emissions could be investigated in future study.

4)      Discussing an outlook on how to maintain and update the database, at which frequency, using which data sources.

[Response] This database will be regularly updated every two or three years, depending on the availability of activity data.

*Specific comments*

-) English could be improved: p.1 l13 "have", l14 "contribute"; p4, l18 "are"; p5, l18: remove "emissions", p7, l9 there are few measurements, p12 l26 "and northward of" needs to be corrected;  p13, l6, "Yevich", p14, l1 "and 5.2%" should be "to 5.2%"; p.17 l13 "publicly"

[Response] They are corrected.

-) abstract: please mention that it is an ANNUAL bottom-up inventory

[Response] It is added.

-) page 2 line 22: is the 2010 number of EPA reported/calculated or projected. If it is the latter,  please make the difference between reported/calculated data and projected data. Also in fig. 2,  make the distinction by have e.g. open circle for projected data.

[Response] The EPA 2010 data are projected. Following your suggestion, we used open circle for the 2010 EPA data in Figure 2.

-) page 3: instead of mentioning "English and Chinese literature", please give the real list of references (and mention the language in the reference list).

[Response] We listed the detailed reference list for each sector in the section 2.

-) page 3: formula: what do you mean exactly with "conditions". Do you mean "technologies/ practices, modi operandi"? Moreover: why is the EF not varying in time but only the correction  factor?

[Response] The conditions have different meanings for different sectors. For example, condition means underground/surface mines for coal mines, practices/managements (organic input, continuous irrigation) for rice paddy, and burning in households/open field

for biomass burning. This word ("conditions") comes from IPCC guidelines vol. 2 (2006). We described the details for EF selections for each sector in the following text. The EF should be varying in time, but little information about time evolution of EF can be available. Thus, we kept the EF unchanged from 1980 to 2010. For the correction factor, normally it correlates with the improvements of technology/practices, or economy development. We can infer the varying correction factor from these indirect indicators. In the revised version, we revised the EF is varying with time in Equation (1), but mentioned that we used constant EFs through the period 1980-2010. "Note that the EFs used in this study did not evolve with time because of limited information about time evolution of EFs." (Page3, line 27)

-) page 4: "CH$_4$ utilization or flaring"? You mean the "CH$_4$ recovery instead of venting into the atmosphere"? Please use the standard reporting language (as also in the CDM)
**[Response]** Yes, it is corrected.

-) page 4 – Table 1: enteric fermentation is (as described in the IPCC GL (2006)) depending for the dairy cattle on the milk production per head and for the non-dairy cattle and other cattle on the live weight per head. These details would be of interest, also complementing the info in the IPCC GL (2006).
**[Response]** We agree that the enteric fermentation depends on the live weight for the non-dairy cattle and milk production for the dairy cattle. In IPCC (2006), the average milk production for the dairy cattle in Asia is 1650 kg/head/yr, and the live weight of non-dairy cattle in China is 300-400 kg/head (Table 10A.2 in IPCC, 2006), which are from the revised 1996 IPCC Guidelines for GHG inventories (IPCC, 1997). Dong et al. (2004) applied the same live weight for non-dairy cattle as IPCC (2006), but higher milk production for the dairy cattle (4000-5000 kg milk /head/yr) for the enteric fermentation, which has a higher EF of female dairy cattle (Table 1). Yamaji et al. (2003), Verburg & Vandergon (2001) and Khalil et al. (1993) referred or adjusted the EFs from previous studies (Dong et al., 2000; Ward and Johnson, 1990; Crutzen et al., 1986) or IPCC guidelines without giving the information of milk production and live weight for the EFs. In the past three decades, the EFs of enteric fermentation could change with milk production for the dairy cattle and live weight for the non-dairy cattle in China. The constant EFs through the past three decades in this study may underestimate the increasing trend of enteric fermentation, because of the increasing milk production per head of dairy cattle and increasing live weight per head non-dairy cattle during the past three decades in China. On the other hand, the increasing weight of crop

production feeding system in livestock and increasing feed of treated straw or residues would reduce the EFs of enteric fermentation (Dong et al., 2004). We discussed the information of milk production and live weight and the uncertainty of possible varying EFs for livestock in the revised version. "Besides the uncertainty of population, the EF of livestock are highly correlated to the live weight per head for meat cattle and milk production per head for dairy cattle (Dong et al., 2004; IPCC, 2006). In this study, as the previous studies, we assumed the EF did not evolve with time because of limited information about the weight distribution of livestock population types besides numbers of animals, although we assessed the uncertainty with different EFs (Table 1). On the one hand, the increasing live weight and milk production per head can increase EFs of enteric fermentation (IPCC, 2006). On the other hand, the increasing weight of crop production feeding system in livestock and increasing feed of treated crop residues can reduce the EFs of enteric fermentation (Dong et al., 2004). The possible changing EF resulting from increased live weight and milk production per head or more feed with treated crop residues should be investigated in a future study." (Page 12, line 5-15)

-) page 5: What do you mean exactly with "biomass burning"? Only small scale or also forest fires, etc. ? Moreover, in formula 2: Why do F and theta not have the index C?

**[Response]** Biomass and biofuel burning includes firewood and crop residues burning as biofuel in rural households, as well as disposed crop residues burning in open field. The $CH_4$ emissions from wildfire of natural ecosystems (forests, grasslands etc.) are not included in our inventory, because this inventory only includes the anthropogenic methane emissions. The total $CH_4$ emissions from wildfire of natural ecosystems in GFED4.1s during 1997 to 2010 is 0.11 Tg $CH_4$/yr, which is less than 5% of $CH_4$ emissions from biofuel burnt in households. In formula 2, theoretically, F and $\theta$ could be different for different crop residues. However, limited information of F and $\theta$ can be available. Thus, we assumed the F and $\theta$ are the same across the six types of crops in formula 2.

-) page 6: in e.g. UK we see huge differences in EF for the fugitive emissions from coal mines, because of different geological underground (based on measurements). Is Zeng et al (2006) for China, a much larger country than UK not reporting a similar large variety?

**[Response]** Zheng et al. summarized regional EFs for six regions (North, Northeast, Northwest, Southwest, Center and South, East of China), which are shown in Table 2. The fugitive EF for coal mines may have large variety across the country because of different coal

bed methane store and different coal mining depth in the coal mines, but is not detailed in Zheng et al. (2006). We used the varied regional average EFs from Zheng et al. (2006) to estimate fugitive emissions from coal mines.

-) page 6: Have emissions estimates from abandoned mines, closed mines been omitted?

**[Response]** On one hand, high moisture in coal strata in China could inundate the abandoned mines, and inhibit the $CH_4$ emissions from abandoned mines. One the other hand, permeability of coalbed in China is small (~0.001 mD), which indicate the limited $CH_4$ emissions from abandoned coal mines. Because 1) the emissions from abandoned mines are less than 1% of total emissions from coal mining (NRDC, 2014), and 2) the time series of numbers and locations of the abandoned mines are unavailable (NRDC, 2014), emissions from abandoned mines are not included in our inventory. This is clarified in the revised version (Page 6, line 27-30).

-) page 6 – Table 2: please specify the $CH_4$ recovery of coal mining gas in the table per province. Please add to the Table also the rice cultivation per province and reflecting as such the difference in cultivated area times the number of cropping seasons. This would be valuable information that adds to the information at Chinese province level in the IPCC GL2006)

**[Response]** The national $CH_4$ recovery of coal mining gas in 1994 and 2000 is reported in Zeng et al. (2006). The database of CDM projects only reported 13 registered coal bed/mine methane before 2009. Thus, we assumed that national average value for $CH_4$ recovery of coal mining gas for each province. For the rice cultivation, the total early, middle and late rice cultivation areas for each province are collected from agriculture statistics yearbooks, and vary year by year. We do not think that it is a good idea to put the yearly varying rice cultivation area of each province in Table 2, otherwise Table 2 is a too "big" to read.

-) page 7 l7: please specify the EFs in kg $CH_4$ per TJ instead of per kton oil or per m3 gas, because the heat value can change significantly between the different types of oil and different types of gas. Please have an evaluation of the gas distribution leakage. Even though Lelieveld et al (2005, Nature) did not found large leakages from transmission pipelines, it is well-known that the gas distribution networks (especially of the old steel pipeline networks in older cities) are subject to large leakages.

**[Response]** For the EFs of fugitive emissions from oil and gas systems, based on the EFs in

UFNCCC (2014) and Schwietzke et al. (2014a, 2014b), in the revised version, we adopted the EFs in Schwietzke et al. (2014a, 2014b) for fugitive emissions from oil and gas systems. For fugitive emissions from oil systems, the average EF is 0.077 kt $CH_4$/PJ (2.9 kg $CH_4/m^3$ oil), and the low and high boundary of EF are 0.058 kt $CH_4$/PJ (2.2 kg $CH_4/m^3$ oil) and 0.190 kt $CH_4$/PJ (7.2 kg $CH_4/m^3$ oil), respectively (see Table 1 in Schwietzke et al., 2014a). These values are consistent with the EFs in the table listed by the reviewer#1. The fugitive $CH_4$ from oil systems increase from 0.36 [0.27-0.98] Tg $CH_4$/yr in 1980 to 0.68 [0.52-1.86] Tg $CH_4$/yr in 2010.

For fugitive emissions from gas systems, we used the fugitive emissions rates (FER) to estimate the fugitive $CH_4$ from gas systems (Schwietzke et al., 2014a, 2014b), including venting, flaring, exploration, production and upgrading, transport, processing, transmission and storage, as well as distribution networks. The gas distribution leakage is included in our inventory. The fugitive emissions rates (FER) of natural gas is decreasing from 1980 to 2011 (Schwietzke et al., 2014b). For China, we adopted the FER linearly decreases from 4.6% (0.81 kt $CH_4$/PJ) in 1980 to 2.0% (0.35 kt $CH_4$/PJ) in 2010. The low and scenario of FER in China decreases from 3.9% in 1980 to 1.8% in 2010, and the high scenario of FER in China decreases from 5.7% in 1980 to 4.9% in 2010. The fugitive $CH_4$ emissions from gas systems increase from 0.45 [0.38-0.56] Tg $CH_4$/yr in 1980 to 1.27 [1.14-3.11] Tg $CH_4$/yr in 2010.

-) page 8, l2: Is the China Env. Stat. Yearbook not showing differences in practices between large versus small or young versus new cities?

**[Response]** No.

-) page 9, l7: please map carefully in a table for each (sub-)sector the specific proxy datasets (over time) are used; page 9, l14 why is livestock distributed with agricultural gross domestic product and GDP and not with the maps of animal numbers, as available from the geonetwork at the FAO site? Why is the oil & gas distributed with GDP, if there are data available on oil and gas exploitation from NOAA? Why considering only 414 coal exploitation sites, if Liu et al (2015, Nature) has a map of several thousand sites. The two-step distribution as described in lines 19-20 should be used for all (sub-)sectors.

**[Response]** We added the details for proxy datasets for the spatial mapping (Table S1). In the revised version, we used gridded maps of animal number as the proxy data for $CH_4$ emissions from livestock, instead of agricultural GDP. For the geographical attribution of

fugitive emissions from oil and gas systems, instead of GDP, we applied the spatial distribution of the EDGARv42 gridded maps for fugitive emissions from oil and gas systems with spatial resolution of 0.1 degree by 0.1 degree, scaled by the total emissions from oil and gas systems in each province (Schwietzke et al., 2014a). The population density, oil and gas production sites, and other proxies for transportation routes are used in EDGARv42 to distribute those $CH_4$ fugitive emissions from oil and gas systems. For coal exploitation, we used data from 414 counties in the previous version, not 414 sites. In each county, there are probably several hundreds of coal production sites. To get a higher spatial resolution, we used the location of 4264 coal production sites from Liu et al. (2015) as proxy data in the revised version. We summed the total annual coal production in each grid of 0.1 degree by 0.1 degree as the weight to distribute the total $CH_4$ emissions from coal mines in each province.

-) page 10, l8: please carefully derive when the acceleration in $CH_4$ emissions start, definitely after 2000, but can we even say in 2002 when China joined the WTO?
**[Response]** We applied piecewise linear regression on the time series of total $CH_4$ emissions, and found that the acceleration in $CH_4$ emissions starts from 2002 (the trends of total $CH_4$ emissions before and after 2002 are 0.5 Tg $CH_4/yr^2$ and 1.3 Tg $CH_4/yr^2$, respectively), which is attributed to the acceleration in $CH_4$ emissions from coal mining after 2002 (the trend of $CH_4$ emissions from coal mining from 2002 to 2010 is 1.1 Tg $CH_4/yr$). This could be related to remarkable achievements in economic index, when China joined WTO starting from December, 2001.

-) page 12, l16: Seen the relative large variation in rice emissions over time (in EDGARv4.2 varying from 19.2 to 11.9 Tg $CH_4/yr$), please compare the emissions of the same years: so the 2005 value of
13.2 Tg $CH_4/yr$ with the NDRC value of 7.9 Tg $CH_4/yr$ and with the Chen (2013) estimate of … in 2005.
**[Response]** It is revised. Note that Chen et al. (2013) compiled all sites data measured in different years and used rice cultivation area in 2008 to estimate the methane emissions from rice cultivation.

-) page 13, l1: maybe a discrepancy can be found in the definition of "biomass burning". Please have a careful look what is included: vegetal waste burning, agricultural waste

burning, crop residue burning, field burning, grassland fires, woodland fires, forest fires, …?

**[Response]** Biomass and biofuel burning includes firewood and crop residues burnt as biofuel in rural households, as well as disposed crop residues burnt in open field. The $CH_4$ emissions from wildfire of natural ecosystems (forests, grasslands etc.) are not included in our inventory, because this inventory only includes the anthropogenic methane emissions. We added the definition of "biomass and biofuel burning" in the method section 2.2.3.

-) page 14: l3: EDGARv4.2 uses the CDM of UNFCCC as input for all developing countries on coal mine gas recovery (cfr. IEA's CO2 from fuel combustion book, part III, GHG).

**[Response]** We also considered the increased utilization of $CH_4$ from coal bed methane (CBM) and coal mine methane (CMM) as accounted for by the increasing utilization fraction in our study (Page 7, lines 1-5), which increased by 4% in the last decade (from 5.2% in 2000 to 9.2% in 2010). Please also see the above response to the $CH_4$ recovery from coal mines.

-) page 16: please give a quantitative evaluation of the mitigation measures and an outlook on the further reduction potential based on the references. Page 16, l6: please evaluate carefully that new PVC gas distribution networks are better than the old steel networks and that new transmission pipelines (such as for the connection Russia and China) are not expected to lead to high leakages. Input on these issues can be gained also from the Chapter 5 of the AMAP report on $CH_4$ from Hoeglund-Isaksson et al. (2016)

**[Response]** Please see the above response to quantitatively estimate the reduction potentials. In this study, we assumed that the fugitive emissions rates (FER) from natural gas systems linearly decreased from 1980 to 2010 because of reduced unintended leakage and technically leakage control. In 2010, the FER in China is 2.0%, including production emissions and the leakage from natural gas production and distribution networks. We agree that PVC pipeline is better than the old steel pipeline. In the 2000s, the networks of natural gas distribution in cities of China are PVC pipelines. If China follow the rates of maximum technically feasible reduction potentials in Höglund-Isaksson et al. (2015), the leakage of long-distance gas transmission could be reduced by 60% and the total fugitive emissions from oil and gas systems can be reduced by 58% in 2030.

**References**

Dong, H., Tao, X., Xin, H. and He, Q.: Comparison of enteric methane emissions in China for

different IPCC estimation methods and production schemes, Trans. Asae, 47(6), 2051–2057, 2004.

IPCC (Intergovernmental Panel on Climate Change): 2006 IPCC Guidelines for National Greenhouse Gas Inventories, IGES, Japan., 2006.

Schwietzke, S., Griffin, W. M., Matthews, H. S., and Bruhwiler, L. M. P.: Global Bottom-Up Fossil Fuel Fugitive Methane and Ethane Emissions Inventory for Atmospheric Modeling, ACS Sustainable Chemistry & Engineering, 2, 1992-2001, 10.1021/sc500163h, 2014a.

Schwietzke, S., Griffin, W. M., Matthews, H. S., and Bruhwiler, L. M. P.: Natural gas fugitive emissions rates constrained by global atmospheric methane and ethane, Environmental Science and Technology, 48, 7714-7722, 10.1021/es501204c, 2014b.

The National Development and Reform Commission (NDRC): The People's Republic of China national Greenhouse gas inventory, 2014. Beijing, China Environmental Press.

Zheng, S., Wang, Y. A. and Wang, Z. Y.: Methane emissions to atmosphere from coal mine in China, Saf. Coal Mines, 36(2), 29–33, 2006. (in Chinese)

---

## Author Comment (AC3) · 28 Aug 2016

Responses to Anonymous Reviewer #3

The paper provides a consistent time series of CH$_4$ emissions from China from 1980-2010. China is an important contributor to total global CH$_4$ emissions and a better understanding of the sources and possible mitigation options is relevant for the scientific community. Methane emission inventories for China have been made before and as such the work is not novel but the compilation of the different sources and the consistent time series make it certainly worthwhile. Also, as discussed in the paper, the discrepancies between various existing estimates for China is substantial and the investigation of the causes or at least identification of sectors that are most uncertain is important for both the global and the Chinese CH$_4$ budget. I think that for several sources the review of emission factors and especially possible trends in these emission factors or the emission controlling variables over the time period could be more in-depth and that this could still further improve the inventory. On the other hand, an inventory includes many sources and a balance between total time spend on each category and the overall result needs to be found. I would recommend the paper for publication but would like to see several points discussed in more detail or added. If for some reason the authors find it unrealistic or over-demanding to make those changes, some argumentation why this is not feasible or out of scope should be provided.

**[Response]** Thank you for your valuable and constructive comments. We revised the manuscript following your suggestions. The details can be found as below.

First of all, as lined out in the beginning of section 2.1., the methods of the IPCC GHG inventory guidelines were followed. The authors then search and use for several, but not all, sources more representative Chinese emission factors. I think it would be valuable to also have a full IPCC emission factor only emission calculation, next to the final result of the paper. This is 1) the easiest way to understand what the impact of the country specific emission factors (EFs) on the total Chinese emission estimate is. 2) In the comparison with the other sources such as EDGAR or EPA – again it would be very useful to know if these estimates' are in line or higher / lower than using avg IPCC EFs. Since the structure followed by the authors is based on the IPCC methodology, my feeling is making an "base-line" avg IPCC EF calculation is not a very demanding task. There are good arguments why the current approach is more accurate but it would provide a very useful benchmark for comparing the impact of more detailed information as well as in the comparison with EDGAR and EPA values.

**[Response]** Following your suggestions, we added the estimates with IPCC default EFs (see Figure 2 and Table S1). In Figure 2, we added the lines for estimates with IPCC EFs and its high/low bound range.

An important aspect of the paper is the long time series. Something that is not well discussed is whether the activity data and emission factor data really cover the temporal changes. For example if

the emission factors are based on using a certain technology but this technology was not used before 1990, the EF might not be representative for the 1980-1990 period. While there are good reasons to use it as best guess, the trend 1980-1990 is then highly uncertain and much less reliable than 1990-2010. I would like to discuss that in more detail for the $CH_4$ emission from rice agriculture.

**[Response]** Thanks. We discussed the uncertainty of changes in EFs on the trend of emissions in the revised version. For the rice paddy sector, the details about the changes in EFs can be found in the next response.

**$CH_4$ emissions from rice agriculture**

In section 2.2.2 the authors explain their approach to calculate $CH_4$ emissions from rice. While it is clearly acknowledged in the paper that the emission factors depend on such things as organic matter (OM) input and water management, no trends in these controlling factors are discussed. Denier van der Gon and Neue (1995) and Denier van der Gon (1999) have provided a simple, empirical impact relationship for $CH_4$ emission from rice fields with OM input versus chemical fertilizer. A ~5 t OM/ha input creates a doubling of the $CH_4$ emission, a 10 t OM/ha triples the $CH_4$ emission. Peng et al use an assumption based on Yan et al (2003) that 50% of the rice paddies received organic input. While that may be the case at a certain moment in time for the trend in $CH_4$ emission it is crucial to understand the trend in the OM input because it is such a strong driver of $CH_4$ emissions from rice fields. Denier van der Gon (1999) compiled the green manure statistics, fertilizer production and harvested rice area statistics in China over the period 1960-1995. Especially from the mid-1970s onwards the production of fertilizer in China grows tremendously but the harvested rice area remains the same or declines somewhat. It is a logical hypothesis that the every year increasing availability of fertilizer (urea) started replacing the much more labor-intensive use of OM incorporation. While reliable statistics for total OM use are lacking, the green manure statistics support this hypothesis. From 1980-1990 the harvest rice area slowly declines, the fertilizer production rapidly increases and the planted green manure area roughly halves. The green manure statistics are available at the regional level and show for example a much stronger impact in the Central and east China (See Figure 3 and table 1 in Denier van der Gon, 1999). The impact of less OM input in the rice field is further enhanced by the change of rice varieties from traditional to high yielding varieties. The main trait of these high yielding varieties is that they are very responsive to N fertilizer and allocate (or invest) a much smaller part of their total net primary production in the below ground root system (which will be the OM for the next growing season). This trend is described by Denier van der Gon (2000) but that paper does not give data for China – nevertheless the high yielding varieties have also been introduced in China and it will also have contributed to making less OM available for $CH_4$ production in Chinese rice soils. This reviewer would therefore argue that the trend for $CH_4$ from rice as shown in table 3 of the paper, strongly underestimated the trend between 1980 and certainly 1990. An educated guess would be that the year 2000 value is realistic and in line with most available estimates as discussed by the authors but the

emissions from rice should show a declining trend in emission since 1980 mostly due to lower OM input into the rice cropping system which is in line with the strong growing availability of urea fertilizer. The authors could use the trends and data compiled in Denier van der Gon 1999 or references therein which would result in a $CH_4$ emission from rice cultivation in an estimated range of ~15 Tg/yr. As a result the trend would be rather similar to EDGAR (Fig 2 in the paper), although the absolute emission level remains lower. Indeed, as mentioned by the authors, the increasing trend in the EDGAR estimate after 2003 is remarkable and not easily understood but that is outside of the scope of the paper.

**[Response]** Thank you for this constructive comment. We fully agree that the decline of rice paddy area with OM input since late of the 1970s can decrease the $CH_4$ emission from rice paddy. In the revised version, we tried to include the estimates of $CH_4$ emission from rice paddy with changing OM input.

In China, OM input includes animal and human wastes, crop straw, green manure and compost and fermented residues. As discussed in Yan et al. (2003), there is little statistic data about the fraction of rice paddy area with OM input as well as the amount of average OM input per hectare (Denier van der Gon, 1999). The only clear message is that the planted area of green manure decrease from 1976 to the late of the 1980s (Denier van der Gon, 1999; China Agricultural Statistic Yearbook), but how much green manure is grown for rice paddy is unclear. Here, several assumptions are applied to get the changes of area of rice paddy with OM input. First, it can be assumed that 100% of rice paddy received OM input before the chemical fertilizer input, since OM input has long history in China (Denier van der Gon, 1999)., Second, we assumed that the area of rice paddy with OM input linearly decreased with the amount of chemical fertilizer input, because most of OM input are labor-intensive and farmers prefer more profitable work in allocating their time rather than preparing OM input for the fields. In the year of 2000, the total chemical fertilizer consumed in China is 41.5 million ton (China Statistic Yearbook, 2001), and 50% of rice paddy with OM input suggested as Yan et al. (2003). Thus, the area of rice paddy with OM input decreased by 1.2% per million ton chemical fertilizer. From 1980 to 2000, the total chemical fertilizer utilization increased from 12.7 million ton to 41.5 million ton (all cultivation types, Figure R1), and the fraction of rice paddy area with OM input decreased from 85% in 1980 to 50% in 2000. After 2000, on one hand, the chemical fertilizer kept increasing (Figure R1); on the other hand, the practice of returning crop residues and using organic fertilizer applications are popularized again because of sustainable quality of arable land and air quality control, which can be indirectly supported by increasing number of the machines for returning crop residues in the 2000s (from 0.44 million in 2004 to 0.62 million in 2011). Thus, in absence of more detailed information we have assumed that the fraction of rice paddy with OM kept stable after 2000. Based on the changing fraction of rice paddy with OM input, $CH_4$ emissions from rice paddy decreased by 3.4 Tg $CH_4$ $yr^{-1}$ (44% of $CH_4$ emission from rice paddy), compared to 1.6 Tg $CH_4$ $yr^{-1}$ with constant fraction of rice paddy with OM input during the period 1980-2010. We used this estimate with the inferred changing fraction of rice paddy with OM input in the revised version, which could correct the underestimated trend of $CH_4$

emission from rice paddy between 1980s and 1990s (Figure R2). Besides the changing OM input, the fraction of rice paddy with continuous irrigation may also changes. But without information of irrigation on rice paddy, we cannot deduce the impact of possible changing irrigation on $CH_4$ emissions from rice paddy. We also discussed the uncertainty from practice of irrigation in the revised version.

[Figure]

Figure R1. The total chemical fertilizer input from 1980 to 2010 in China.

[Figure]

Figure R2. $CH_4$ emissions from rice cultivation in the previous and revised version.

**CH₄ Emissions from Oil and Gas industry**

Emissions from natural gas production sites are characterized by skewed distributions, where a small percentage of sites—commonly labeled super-emitters—account for a majority of emissions. (Zavala-Araiza et al., 2015). The importance of these super-emitters in the O&G sector is a rather new insight and probably not well represented in the current emission factors. It only surfaced due to large numbers of measurements that showed the "fat tail distribution" of the EFs. Therefore, I would argue that using standard emission factors may well lead to underestimation for the emissions from this sector. Moreover, the emission factors used in the paper appear really low. I would like to see a very simple "sanity check" on these numbers. When taking the total calculated CH₄ emission from the Oil and Gas industry for example in 2000 (0.1 Tg / yr) or 2010 (0.3 Tg/yr) (see Table 3 in the paper); what share is due to the gas industry and what percentage of total natural gas production is this? And does it make sense over time? At a first glance it seems a really low estimate that is presented here. To get a feeling I have taken the data from Schwietzke et al., 2015 and looked at the CH₄ emissions from china from Natural gas industry only if a Fugitive Emission Rate (FER) of 1% is assumed (see figure below). This leads to a factor 2 higher emissions than reported by Peng et al. and the gap is much bigger because in the below estimate oil industry is not included whereas Peng's estimate includes both oil and gas. While this does not mean that the presented estimated in the paper is wrong, I would like to see more discussion and think that expressing the FER as a % of the production is a very useful thing to do to show that really low % are currently assumed in this paper whereas recent measurements in the US and Canada found FER's of 2-4% more realistic.

Constant global avg. Fugitive Emission Release (FER) of 1% of natural gas production only: data taken from Schwietzke et al., 2014. The figure does not include the oil sector emissions yet but these are available from Schwietzke et al and would further increase the emission estimate.

[Figure]

**[Response]** Thank you for pointing out this possible underestimated EFs for fugitive emissions from oil and natural gas systems. Comparing the default EFs of IPCC (2006) and EFs in Schwietzke et al. (2014a, 2014b), the EFs from Zhang and Chen (2010) and NDRC (2014) in the previous version have smaller values. Considering the EFs in USA, Canada and other countries from UNFCCC (2014) and Schwietzke et al. (2014a, 2014b), in the revised version, we adopted the EFs in Schwietzke et al. (2014a, 2014b) for fugitive $CH_4$ from oil and natural gas systems (see reply to the reviewer#1). For fugitive emissions from oil systems, the average EF is 0.077 kt $CH_4$/PJ (2.9 kg $CH_4$/m$^3$ oil), and the low and high boundary of EF are 0.058 kt $CH_4$/PJ (2.2 kg $CH_4$/m$^3$ oil) and 0.190 kt $CH_4$/PJ (7.2 kg $CH_4$/m$^3$ oil), respectively (see Table 1 in Schwietzke et al., 2014a). These values are consistent with the EFs in the table listed by the reviewer#1. The fugitive $CH_4$ from oil systems increase from 0.36 [0.27-0.98] Tg $CH_4$/yr in 1980 to 0.68 [0.52-1.86] Tg $CH_4$/yr in 2010.

For fugitive $CH_4$ from natural gas systems, the fugitive emissions rates (FER) of natural gas is decreasing from 1980 to 2011 (Schwietzke et al., 2014b). For China, We assumed a FER linear decrease from 4.6% (0.81 kt $CH_4$/PJ) in 1980 to 2.0% (0.35 kt $CH_4$/PJ) in 2010, which is today close to the FER (1.9%) in OECD countries in 2010. The range of uncertainty was estimated with a scenario assuming a low FER in China decreasing from 3.9% in 1980 to 1.8% in 2010, and a scenario with high FER in China decreasing from 5.7% in 1980 to 4.9% in 2010. The fugitive $CH_4$ from natural gas systems increased from 0.45 [0.38-0.56] Tg $CH_4$/yr in 1980 to 1.27 [1.14-3.11] Tg $CH_4$/yr in 2010.

The total $CH_4$ emissions from oil and natural gas systems increase from 0.81 [0.65-1.54] Tg $CH_4$/yr in 1980 to 1.95 [1.66-4.98] Tg $CH_4$/yr in 2010, which is consistent with the values from Schwietzke et al. (2014a) and is lower than EDGARv42, but higher than the values reported by NDRC (2014). In the revised version, we also applied the spatial distribution of EDGARv42 grid maps with spatial resolution of 0.1 degree by 0.1 degree, scaled by the total emissions from oil and gas systems in each province (Schwietzke et al., 2014a), which could have better geographical distribution than the GDP proxy used in the previous version.

**References**

Denier van der Gon, H.A.C., Neue, H.U., Influence of organic matter incorporation on the methane emission from a wetland rice field, Global Biogeochemical Cycles 9, pp. 11-22, 1995.

Denier van der Gon, H.A.C., Changes in CH$_4$ Emissions from Rice Fields from 1960 to the 1990s. II. The declining use of organic inputs in rice farming, Global Biogeochemical Cycles 13, 1053-1062, 1999.

Denier van der Gon, H.A.C., Changes in CH$_4$ Emissions from Rice Fields from 1960 to the 1990s. I. Impacts of modern rice technology, Global Biogeochemical Cycles 14, 61-72, 2000.

Sass, R.L., F.M. Fisher, F.T. Turner and M.F. Jund, (1991) Methane Emission from Fice Fields as Influenced by Solar Radiation, Temperature and Straw Incorporation, Global Biogeochem. Cycles, 5, 335-350.

Schwietzke, S., Griffin, W. M., Matthews, H. S., and Bruhwiler, L. M. P.: Global Bottom-Up Fossil Fuel Fugitive Methane and Ethane Emissions Inventory for Atmospheric Modeling, ACS Sustainable Chemistry & Engineering, 2, 1992-2001, 10.1021/sc500163h, 2014a.

Schwietzke, S., Griffin, W. M., Matthews, H. S., and Bruhwiler, L. M. P.: Natural gas fugitive emissions rates constrained by global atmospheric methane and ethane, Environmental Science and Technology, 48, 7714-7722, 10.1021/es501204c, 2014b.

IPCC (Intergovernmental Panel on Climate Change): 2006 IPCC Guidelines for National Greenhouse Gas Inventories, IGES, Japan. 2006.

The National Development and Reform Commission (NDRC): The People's Republic of China national Greenhouse gas inventory, 2014. Beijing, China Environmental Press.

Yan, X. Y., Ohara, T. and Akimoto, H.: Development of region-specific emission factors and estimation of methane emission from rice fields in the East, Southeast and South Asian countries, Glob. Chang. Biol., 9(2), 237–254, doi:10.1046/j.1365-2486.2003.00564.x, 2003.

Zavala-Araiza, D.; Lyon, D.; Alvarez, R. A.; Palacios, V.; Harriss, R.; Lan, X.; Talbot, R.; Hamburg, S. P. Toward a functional definition of methane super-emitters: Application to natural gas production sites. Environ. Sci. Technol. 2015, 49, 8167–8174.

---

## Referee Report (RR1)

**Review of revised paper of Peng et al. (2016) "Inventory of anthropogenic methane emissions in Mainland China from 1980-2010.**

**18 September 2016**

**Correspondence to Anonymous Reviewer #2.**

**Upfront note**

The authors did improve the paper on the spatial distribution, documenting in section 2.3 and Table S1 the method and proxy datasets used. I would still have expected some recommendation on the seasonality. The latter does not have to be a full set of monthly profiles, but an evaluation of what is so far used in chemical transport models and its representativeness (e.g. taking into account certain habits, frequency of rice yield per year, etc.).

**General comments**

1) The authors addressed the question of abandoned mines fully. Can the authors also add something with regard to the coal fires (IPCC category 7)?
2) The authors addressed in full the spatial proxy data in section 2.3 and Table S1.
3) The authors should not only take into account the second communication of China to the UNFCCC with the 2005 inventory but also the first communication with the inventory of 1994 (in the middle of the 31 years which the paper addresses). Moreover the authors should include these two points in Fig. 2. For section 4.1 I also would recommend to enlarge the comparison description in the first 5 lines with the first national communication. For the last lines of this first paragraph, I would recommend to clarify the last part of the sentence ("is under investigation by the EDGARv4.2 team") , given that this investigation since 2013 (when Bergamaschi noted this slope error) meanwhile might have been concluded.

**Concerning the enriching of the paper content**

For the points 1), 3) and 4) it would be worthwhile that the authors provide in the introduction of their paper an indication of what the reader can expect and what not. This could include recommendations for the temporal distribution, the frequency of the inventory update, and an outlook on the reduction potentials (indicating the sectors of interest for a future study).

Concerning point 2), for the sake of clarity, it would be nice to repeat in the introduction the definition of "Mainland China" for the full period. That Taiwan islands are not included is only part of it, Macao (as established nowadays) differs from the years before 1999 in status. It is important for inventory compilers to know exactly which regions/provinces the activity data include for the full period (also in the period before 1999).

**Specific comments**

These were addressed in a satisfactory way.

---

## Author Response (AR2)

**Response to review of revised paper of Peng et al. (2016) "Inventory of anthropogenic methane emissions in Mainland China from 1980-2010.**

**Upfront note**

The authors did improve the paper on the spatial distribution, documenting in section 2.3 and Table S1 the method and proxy datasets used. I would still have expected some recommendation on the seasonality. The latter does not have to be a full set of monthly profiles, but an evaluation of what is so far used in chemical transport models and its representativeness (e.g. taking into account certain habits, frequency of rice yield per

10 year, etc.).

[Response] Thanks. The main purpose of this study is representing annual methane inventory of China. The development of seasonality require quite a lot of work, and the confidence of seasonality of $CH_4$ emissions cannot be guaranteed as the annual product. We agree that the seasonality is important, and evaluation of that used in chemical transport models is essential. But this is beyond the scope of this study. The seasonality of $CH_4$ emissions is

15 worth a study in itself as it is mostly not even done in global inventories. We hope that it can be addressed in future study.

**General comments**

20 1) The authors addressed the question of abandoned mines fully. Can the authors also add something with regard to the coal fires (IPCC category 7)?

[Response] In our inventory, emissions from underground coal fires (IPCC category 7) are not included, because 1) it is unclear how much coal is yearly burnt by underground coal fires from 1980 to 2010, and 2) less than 0.01 $Tg\ CH_4\ yr^{-1}$ (less than 0.1% of total emissions from coal mining) are emitted from underground coal fires during

25 the 2000s (EDGARv4.2). We clarified this in the revised version.

2) The authors addressed in full the spatial proxy data in section 2.3 and Table S1.

3) The authors should not only take into account the second communication of China to the UNFCCC with the

30 2005 inventory but also the first communication with the inventory of 1994 (in the middle of the 31 years which the paper addresses). Moreover the authors should include these two points in Fig. 2. For section 4.1 I also would recommend to enlarge the comparison description in the first 5 lines with the first national communication. For the last lines of this first paragraph, I would recommend to clarify the last part of the sentence ("is under investigation by the EDGARv4.2 team"), given that this investigation since 2013 (when

35 Bergamaschi noted this slope error) meanwhile might have been concluded.

[Response] Thanks. We added the values from the first and second National Communication Climate Change of China to the UNFCCC in Fig. 2 (diamond marker), as well as in the text. For the investigation of higher increase of

total CH$_4$ emissions in China by EDGAR team, the CH$_4$ emissions from coal mines in the next release of EDGAR are lower than EDGARv4.2 (G. Maenhout, pers. Comm.). The sentence is revised as "**The slower increase of total CH$_4$ emissions in China than reported by EDGARv4.2 has already be noticed (e.g. Bergamaschi et al., 2013; Saunois et al., 2016) and is improved in the new EDGARv4.3.2 release, in which the total fugitive emissions from coal mining in China is 1.6 times lower than EDGARv4.2 and distributed over about 20 times more point source locations. Lin (2016) assessed the EDGARv4.2 and EDGARv4.3.2 coal mine emissions within her inverse modeling study and showed lower coal mine emissions than EDGARv4.2 over Asia**." in the revised version.

**Concerning the enriching of the paper content**

For the points 1), 3) and 4) it would be worthwhile that the authors provide in the introduction of their paper an indication of what the reader can expect and what not. This could include recommendations for the temporal distribution, the frequency of the inventory update, and an outlook on the reduction potentials (indicating the sectors of interest for a future study).

**[Response]** Thanks. We added the information you suggested in the introduction.

Concerning point 2), for the sake of clarity, it would be nice to repeat in the introduction the definition of "Mainland China" for the full period. That Taiwan islands are not included is only part of it, Macao (as established nowadays) differs from the years before 1999 in status. It is important for inventory compilers to know exactly which regions/provinces the activity data include for the full period (also in the period before 1999).

[revised manuscript text omitted]

---

## Author Response (AR3)

**Response to review of revised paper of Peng et al. (2016) "Inventory of anthropogenic methane emissions in Mainland China from 1980-2010.**

Comments to the Author:

5 Although a full analysis of monthly emissions is not conducted in this study, some recommendations on the seasonality should be provided in the text.

I thinks the paper is ready for publication after addressing this issue.

10 **[Response]** Thanks. We added some recommendations on the seasonality in the text: "
[revised manuscript text omitted]